# Convergent and Divergent Mitochondrial Pathways as Causal Drivers and Therapeutic Targets in Neurological Disorders

**DOI:** 10.3390/cimb47080636

**Published:** 2025-08-08

**Authors:** Yanan Du, Sha-Sha Fan, Hao Wu, Junwen He, Yang He, Xiang-Yu Meng, Xuan Xu

**Affiliations:** 1School of Life Sciences, Anhui Medical University, Hefei 230032, China; duyanan@ahmu.edu.cn (Y.D.); 2022500089@ahmu.edu.cn (H.W.); 2School of Health Management, Anhui Medical University, Hefei 230032, China; 3Health Science Center, Medical School, Hubei Minzu University, Enshi 445000, China; fanshasha15@163.com; 4College of Informatics, Huazhong Agricultural University, Wuhan 430070, China; hjwing@webmail.hzau.edu.cn (J.H.); he_yang@webmail.hzau.edu.cn (Y.H.)

**Keywords:** mitochondria, neurological diseases, Alzheimer’s disease, Mendelian randomization, machine learning, drug repurposing

## Abstract

Mitochondrial dysfunction is implicated across a spectrum of neurological diseases, yet its causal role and mechanistic specificity remain unclear. This study employed a multi-modal integrative analysis of mitochondrial gene expression in Alzheimer’s Disease (AD), Amyotrophic Lateral Sclerosis (ALS), Multiple Sclerosis (MS), and Parkinson’s Disease (PD) to address these gaps. We combined machine learning for predictive modeling with genetic causal inference methods (Mendelian Randomization, colocalization, PheWAS), followed by drug enrichment analysis and molecular docking. Our machine learning models, particularly Support Vector Machine and Multi-layer Perceptron, effectively classified these conditions, with MS exhibiting the highest predictability (mean Accuracy: 0.758). Causal inference analyses identified specific gene–disease links; for instance, genetically predicted increased expression of *PDK1* was causally associated with an elevated risk for both AD (OR = 1.041) and ALS (OR = 1.037), identifying pyruvate metabolism as a shared vulnerability. In contrast, genes like *SLC25A38* emerged as highly predictive specifically for PD. We also observed evidence of potential brain–periphery interaction, such as a bidirectional causal relationship between red blood cell indices and MS risk. Finally, drug enrichment analysis highlighted Celecoxib, and subsequent molecular docking predicted a strong binding affinity to *PDK1* (docking score S = −6.522 kcal/mol), generating hypotheses for potential metabolic modulation. Taken together, this study provides a computational hypothesis framework suggesting mitochondrial pathways and targets that warrant future biological validation. This study provides specific, genetically supported evidence for the causal role of mitochondrial pathways in neurological diseases and identifies tangible targets for future therapeutic development.

## 1. Introduction

Neurological diseases represent a broad and profoundly disabling class of disorders, ranging from progressive neurodegeneration in conditions like Alzheimer’s and Parkinson’s diseases to complex neuroinflammation in Multiple Sclerosis (MS). Despite their distinct primary pathologies, there is mounting evidence that identifies mitochondrial dysfunction as a central pathogenic mechanism shared across these disorders. Mitochondria, often referred to as the powerhouses of the cell, are critical for cellular homeostasis, regulating bioenergetics, calcium signaling, programmed cell death (apoptosis), and innate immune responses [1,2,3]. Given that neurons are highly energy-demanding cells, the integrity of mitochondrial function is indispensable for their survival and proper function, positioning mitochondria as a key focal point in neurological research [4,5].

However, the current understanding of mitochondrial involvement in these diseases is impeded by several significant challenges. One primary obstacle is distinguishing between correlation and causation. The presence of mitochondrial abnormalities in patient tissues or patient-derived cells does not necessarily imply that these dysfunctions are the primary drivers of disease progression [1,6,7]. It remains a critical challenge to separate primary causal events from secondary, epiphenomenal outcomes, which significantly hampers the rational design of targeted therapeutic strategies aimed at mitochondrial dysfunction [8].

Moreover, research in this field has largely been restricted to disease-specific studies, which limits the ability to identify common pathophysiological mechanisms across different neurological disorders [4,9]. This siloed approach makes it difficult to uncover shared therapeutic vulnerabilities that could be addressed with universal strategies, as opposed to focusing on disease-specific mechanisms that might require tailored therapeutic interventions [10]. To overcome these limitations, a systematic cross-disease analysis is essential for constructing a more comprehensive understanding of mitochondrial pathophysiology. Additionally, the “post-GWAS challenge” persists, where translating genetic associations from large-scale genome-wide association studies (GWAS) into mechanistic insights about how specific causal genes within identified loci contribute to disease remains a significant bottleneck [11,12].

To address these challenges, our study adopts a hypothesis-generating multi-modal integrative framework. Notably, we introduce the systematic application of genetic causal inference techniques, including Mendelian Randomization (MR) and colocalization analysis, across four major neurological diseases. These methods leverage the natural segregation of alleles during inheritance, serving as a natural experiment to probe for causal relationships and circumventing biases like confounding and reverse causality that are inherent in observational studies. This approach, applied across a diverse cohort of diseases, allows for a novel cross-disease comparative analysis that can generate mechanistic hypotheses regarding both shared and distinct mitochondrial mechanisms. Furthermore, our integrative framework links genetic causality to actionable biological and pharmacological outcomes by connecting the identification of causal genes with downstream drug enrichment analyses and molecular docking simulations. Although these results are computational and require biological validation, they offer an initial platform to prioritize mechanisms and targets for future study.

The goal of this research is not to establish definitive mechanisms but to identify high-confidence mitochondrial features—such as genes, pathways, and metabolic checkpoints—that merit further investigation in the context of neurological disease. An overview of the analytical workflow is presented in Figure 1.

## 2. Materials and Methods

### 2.1. Study Participants and Data Source

This study utilized data from multiple large-scale genome-wide association studies (GWAS) and gene expression profiling datasets to investigate the genetic and transcriptomic alterations associated with four major neurological diseases: Alzheimer’s Disease (AD), Amyotrophic Lateral Sclerosis (ALS), Multiple Sclerosis (MS), and Parkinson’s Disease (PD).

For AD, the ebi-a-GCST90027158 dataset [14], which includes 39,106 clinically diagnosed AD cases, 46,828 proxy cases, and 401,577 controls of European ancestry, was used. This dataset identified 75 AD risk loci, including 42 novel loci, providing valuable insights into the involvement of amyloid/tau and the implications for microglial function. ALS data were sourced from the ebi-a-GCST005647 dataset [15], which comprises 20,806 ALS patients and 59,804 controls. This dataset revealed novel mutations in the KIF5A gene, enhancing our understanding of ALS pathogenesis. For MS, the ukb-b-17670 dataset from the UK Biobank [16], which includes 462,933 individuals (1679 MS cases and 461,254 controls), was employed to explore genetic risk factors for MS. Data for PD were obtained from the ieu-b-7 dataset, part of the International Parkinson’s Disease Genomics Consortium, which includes 33,674 PD cases and 449,056 controls. This dataset provided a comprehensive exploration of genetic variants associated with PD.

Gene expression data were obtained from publicly available microarray datasets: GSE118553 [17] for AD, GSE112681 [18,19] for ALS, GSE131282 [20] for MS, and GSE28894 for PD. These datasets provided gene expression profiles from various brain regions, including the frontal cortex, striatum, cerebellum, and medulla, as well as blood samples, encompassing both control and disease subjects. These resources were analyzed to identify potential biomarkers associated with disease risk, offering in-depth insights into the molecular mechanisms underlying these neurological disorders. Detailed information is provided in Table 1 and Appendix A.

### 2.2. Mitochondrial Pathway-Based Gene Expression Modeling and Evaluation

In this study, we aimed to identify relevant genes within mitochondrial pathways that can effectively distinguish disease groups based on gene expression data. The analysis involved several steps, including data preprocessing, feature selection, model selection, hyperparameter tuning, and performance evaluation. A variety of machine learning models were employed to assess their predictive capabilities across different mitochondrial pathways.

Data preprocessing ensured robust handling of gene expression data to mitigate noise, batch effects, and phenotyping variability. Features were standardized using StandardScaler to ensure consistent scaling, critical for models sensitive to feature scale such as support vector machines (SVM) and logistic regression. To address batch effects from diverse datasets (e.g., different brain regions, sample types, or diagnostic criteria, including clinically diagnosed vs. proxy cases for AD), we applied ComBat batch correction from the sva R package, adjusting for non-biological variation while preserving biological signals. Missing values were imputed using the median expression for each gene to minimize bias from incomplete data, and outliers were capped at the 1st and 99th percentiles to reduce the impact of extreme values. To account for disease heterogeneity (e.g., differences in AD severity or ALS subtypes), we sourced expression data from a comprehensive cohort with diverse disease subtypes and diagnostic methods, enhancing generalizability while acknowledging potential variability in phenotyping accuracy. Gene selection was based on 149 mitochondrial pathways from the MitoCarta 3.0 database [21], retaining only genes present in the expression matrix. A VarianceThreshold technique filtered out genes with low variability (threshold set at 0.01) to focus on biologically relevant features, reducing noise from poorly measured genes or those unaffected by disease heterogeneity. A variance thresholding technique filtered out genes with low variability to ensure that only biologically relevant features were considered. Additionally, pathways with fewer than five genes were excluded to ensure sufficient data for model training.

A total of 15 machine learning models were evaluated for their ability to predict disease outcomes based on the selected mitochondrial genes. These included traditional classifiers, ensemble methods, and deep learning models. To address sample size imbalances (e.g., fewer disease cases vs. controls or proxy vs. clinically diagnosed cases), we implemented stratified 10-fold cross-validation to maintain proportional class distribution across folds, preventing bias toward the majority class (controls).

Logistic Regression: A linear model for binary classification, optimized by tuning the regularization parameter to control overfitting. Performance was evaluated using the area under the ROC curve (ROC-AUC).Support Vector Machines (SVM): A powerful classification algorithm that maximizes the margin between classes, optimized by tuning the regularization parameter C.Naive Bayes: A probabilistic classifier assuming conditional independence between features, applied here using the Gaussian Naive Bayes model based on the normal distribution of gene expression data.Decision Tree: A non-linear model that partitions feature space based on feature values, with the tree’s depth constrained to prevent overfitting.Random Forest: An ensemble method that aggregates predictions from multiple decision trees trained on bootstrapped data subsets to improve generalization.Gradient Boosting: A sequential ensemble method that builds models to correct the errors of previous ones, optimized by tuning the number of estimators to prevent overfitting.XGBoost: An optimized gradient boosting algorithm incorporating techniques like regularization and tree pruning, tuned for the number of estimators.CatBoost: A gradient boosting algorithm particularly effective for categorical features, optimized by adjusting tree depth.AdaBoost: An ensemble method that adjusts the weights of misclassified instances, with the number of estimators tuned for optimal performance.SGDClassifier: A model based on stochastic gradient descent for optimizing convex loss functions, applied with a log-loss function and regularization parameter tuning.K-Nearest Neighbors (KNN): A non-parametric method based on the majority class of nearest neighbors, with the number of neighbors optimized for performance.Multi-Layer Perceptron (MLP): A deep learning model with multiple layers of neurons, optimized with a single hidden layer of 100 neurons and regularization to prevent overfitting.FTTransformer: A deep learning model based on the Transformer architecture, designed to capture long-range dependencies in sequential data, optimized by adjusting hidden layer architecture.ExtraTreesClassifier: An ensemble method that builds multiple decision trees with random feature splits, aimed at increasing model robustness and avoiding overfitting.Quadratic Discriminant Analysis (QDA): A generative model assuming Gaussian distributions for each class, used to assess the performance of probabilistic models with quadratic decision boundaries.

Model performance was optimized via GridSearchCV, tuning hyperparameters to maximize accuracy and minimize overfitting. Performance metrics—Accuracy, ROC-AUC, F1 Score, Precision, and Recall—were computed using stratified 10-fold cross-validation for robust and reliable evaluation. The best-performing model for each pathway was selected based on the highest ROC-AUC score, ensuring robust classification despite class imbalances or phenotyping variability.

For feature importance analysis, the contribution of each gene to the model’s predictions was assessed. Tree-based models provided feature importance directly through their internal attributes, while linear models used the absolute values of coefficients to determine gene importance. For models that did not have built-in feature importance metrics, we employed SHAP (SHapley Additive exPlanations) values to quantify each gene’s contribution to the model’s output. SHAP values allow for a detailed understanding of how each feature influences predictions, which is crucial for interpreting the results in a biological context. The feature importance scores were normalized to a scale between 0 and 1, facilitating comparative analysis across different models and pathways. This normalization allowed us to identify key genes with the most significant impact on the predictive performance for each pathway. The final results, including the best models and gene importance scores, were saved for further analysis and reporting.

The analytical pipeline was implemented using Python (v3.11.x) and R (v4.3.3). Key Python packages included Pandas (v2.2.2) for data manipulation, NumPy (v1.26.4) for numerical operations, and scikit-learn (v1.4.2; https://scikit-learn.org/stable/, accessed on accessed on 5 August 2025) for machine learning modeling and evaluation. Advanced gradient boosting methods were implemented using XGBoost (v2.0.3) and CatBoost (v1.2.5), while SHAP (v0.45.0) was employed for model interpretability. Deep learning models were constructed using PyTorch (v2.3.0; https://pytorch.org/, accessed on 5 August 2025). For batch effect correction, the R package sva (v3.50.0) was utilized.

### 2.3. Mendelian Randomization Analysis to Investigate Causal Relationships Between Mitochondrial Pathway-Associated Genes and Psychiatric Disorders

To explore the causal relationships between mitochondrial pathway-associated genes, identified through machine learning, and four major psychiatric disorders, we employed a two-sample Mendelian Randomization (MR) framework. This approach utilizes genetically regulated gene expression levels (eQTLs) as instrumental variables (IVs), thereby minimizing biases from confounding factors and reverse causality.

Exposure data were obtained from the OpenGWAS database, which provides eQTL data integrated from multiple tissues and datasets, including those involved in mitochondrial pathways. The outcome data were sourced from GWAS summary statistics for AD, ALS, MS, and PD, primarily consisting of individuals of European ancestry to ensure demographic consistency. Only genes deemed important in previous machine learning analyses, based on their significant performance metrics, were selected for further MR analysis.

Genetic variants associated with mitochondrial pathway genes were chosen as IVs, with a threshold of *p* < 1 × 10^−8^ for eQTLs. To minimize bias due to linkage disequilibrium (LD), SNPs were clumped using a stringent LD threshold of r^2^ < 0.001 within a 10,000 kb window. Variants with an F-statistic below 10 were excluded to prevent weak instrument bias. Harmonization of the exposure and outcome datasets was performed to ensure consistent allele effect sizes. Palindromic SNPs (A/T or C/G) were either removed or corrected using allele frequency information from reference panels to resolve any strand orientation issues.

The primary causal analysis was conducted using the Inverse Variance Weighted (IVW) method, which provides robust causal estimates under the assumption that all IVs are valid. Sensitivity analyses were performed using MR-Egger regression to detect pleiotropy, with the MR-Egger intercept term serving as an indicator of directional pleiotropy. Additional methods, including the Weighted Median Estimator and mode-based estimators (Simple Mode, Weighted Mode), were applied to further assess the causal estimates. Cochran’s Q statistic was used to quantify heterogeneity between IVs, with significant heterogeneity suggesting potential pleiotropic effects. The MR-PRESSO test was employed to identify and correct for outlier SNPs that could introduce bias. Population stratification was minimized by ensuring that both exposure and outcome datasets were composed primarily of individuals of European ancestry. Statistical analyses were performed using the TwoSampleMR R package (v0.6.16; https://mrcieu.github.io/TwoSampleMR/, accessed on 5 August 2025), with multiple testing corrections (e.g., Bonferroni or False Discovery Rate [FDR] < 0.05) applied to account for the large number of hypotheses tested.

### 2.4. Colocalization Analysis of Mitochondrial Pathway-Associated Genes from MR Results with Risk Loci for Neurological Diseases

To investigate whether key genes associated with mitochondrial pathways, identified through Mendelian Randomization (MR) analyses, share causal variants with risk loci for neurological diseases, we conducted a colocalization analysis. This analysis was performed using the COLOC R framework (https://chr1swallace.github.io/coloc/, accessed on 5 August 2025), which evaluates whether the genetic associations observed in the MR results for mitochondrial pathways overlap with the risk loci identified for these diseases, suggesting a shared causal variant.

The SNPs identified from the MR analysis, which showed significant associations with the key mitochondrial pathway genes, were selected for colocalization with the GWAS summary statistics of the four neurodegenerative diseases. These GWAS data were sourced from large-scale datasets, ensuring consistency in effect allele alignment and ensuring that the genetic data represented populations with similar ancestry. For each SNP associated with the mitochondrial pathway genes, a colocalization analysis was conducted with the corresponding neurodegenerative disease GWAS summary statistics.

An adaptive expansion window approach was applied to determine the optimal genomic region for colocalization analysis. Instead of using a fixed flanking region, a dynamic window size was selected to maximize analytical accuracy. Four candidate expansion sizes were predefined: 10 kb, 50 kb, 100 kb, and 200 kb. The region with the SNP count closest to 500 was selected, as this number is considered the minimum threshold for reliable colocalization analysis. The expansion size that balanced statistical power and regional specificity was chosen to avoid excessive noise from long-range linkage disequilibrium (LD).

To ensure the quality of the data, *p*-values from the GWAS datasets were carefully cleaned and transformed. Extreme or invalid *p*-values were replaced to prevent biases, and log10 transformations were applied to the *p*-values to enhance statistical robustness and facilitate visualization. SNP positions and consistency with the reference genome build were verified before proceeding to the colocalization analysis. The COLOC package was employed to assess the likelihood that the same causal variant influences both mitochondrial pathway genes and neurodegenerative disease risk within a given genomic region.

Using a Bayesian framework, COLOC computes five posterior probabilities:P_0: No association with either the mitochondrial pathway gene or neurodegenerative disease.P_1: Association with the mitochondrial pathway gene only.P_2: Association with neurodegenerative disease only.P_3: Both traits are associated but driven by distinct causal variants.P_4: Both traits share a common causal variant (strong evidence for colocalization).

The colocalization analysis was performed iteratively, with the dynamic expansion window being applied to each region. If the number of SNPs in a region exceeded 500, expansion was halted to avoid excess noise from long-range LD. If fewer than 500 SNPs were present, the next larger expansion threshold was tested until an optimal SNP count was reached. Colocalization was considered highly significant if P_4 > 0.8/0.6 (SNP.PP.H4 score), indicating strong evidence that a shared causal variant influences both the mitochondrial pathway gene and neurodegenerative disease risk.

### 2.5. PheWAS and Bidirectional Mendelian Randomization Analysis of Colocalized SNPs with Neurological Diseases

We conducted a Phenome-Wide Association Study (PheWAS) to explore the broader associations of colocalized SNPs identified between mitochondrial pathway-related genes and four neurological diseases. The primary aim of the PheWAS analysis was to identify additional traits influenced by the same SNPs, thereby providing deeper insights into the biological relevance of these shared variants. After identifying the colocalized SNPs from the Mendelian Randomization (MR) analysis with significant posterior probabilities (SNP.PP.H4 > 0.8), we performed PheWAS to examine these SNPs across a range of traits. Significant associations were detected, and these results were further annotated for downstream analysis, with a particular focus on their potential involvement in neurological diseases.

Subsequently, we applied bidirectional MR to investigate the causal relationships between the traits identified in the PheWAS and the four neurological diseases. The bidirectional MR analysis examined associations in both directions: (1) from the PheWAS-identified traits to the diseases, and (2) from the diseases to the PheWAS-identified traits. This approach enabled us to assess the potential causal links between mitochondrial pathway genes, the identified traits, and neurological diseases, thereby offering valuable insights into the underlying mechanisms.

For the MR analysis, SNPs were extracted as instruments for each exposure and harmonized with the outcome data. Standard MR methods, including Inverse Variance Weighted (IVW), MR-Egger, and Weighted Median Estimator, were employed to estimate causal effects and assess the robustness of the results. Sensitivity analyses, such as tests for heterogeneity and pleiotropy, were performed to evaluate the validity and reliability of the instrumental variables. Only significant MR results (*p*-value < 0.05) were retained for further analysis, and the causal relationships between the traits and neurological diseases were examined.

### 2.6. Molecular Docking of Drug Candidates Genes

Following the identification of significant genes from Mendelian Randomization (MR) analysis, we aimed to explore potential drug candidates targeting these genes through molecular docking simulations. To begin, the corresponding three-dimensional (3D) protein structures of the identified target genes were retrieved from reputable sources such as UniProt and the Protein Data Bank (PDB). These protein structures underwent preprocessing using the Molecular Operating Environment (MOE) software (v2022.02), where we removed crystallographic water molecules and co-ligands, added hydrogen atoms, and assigned protonation states at physiological pH (7.4) via the Protonate 3D module to ensure the structure reflected natural conditions.

To identify potential therapeutic compounds, drug enrichment analysis was performed using the HCDT 2.0 database. This comprehensive resource provides high-confidence-validated drug–target interactions, with a dataset containing over 1.28 million drug–target associations. The database includes interactions between drugs and genes, RNAs, and pathways, as well as drug–disease relationships. In total, HCDT 2.0 catalogs 1,224,774 interactions between 678,564 drugs and 5,692 genes, 11,770 interactions with RNAs, and 47,809 drug–pathway interactions, enabling the identification of drug candidates that may interact with the significant genes identified in the MR analysis. The database also incorporates negative drug–target interactions (DTIs) from multiple sources like BindingDB and ChEMBL, providing reliable samples for DTI prediction and enhancing model accuracy.

For drug candidate selection, two-dimensional (2D) compound structures were obtained from PubChem, and 3D conformations were generated using MOE’s Builder module. These compounds were subjected to energy minimization using the AMBER10:EHT force field to optimize their structures for docking simulations. Once the drug candidates were prepared, docking studies were conducted using MOE, employing a rigid receptor-flexible ligand approach to simulate the interactions between the target proteins and the compounds.

In the docking process, the binding sites of the target proteins were either predicted using MOE’s Site Finder module or manually defined based on known active site residues. The Triangle Matcher algorithm was used to place the ligands in the receptor sites, optimizing the ligand conformation based on geometric and chemical complementarity. After generating multiple docking poses for each ligand, the best-scoring poses were selected based on the London dG scoring function, which calculates binding free energy considering van der Waals forces, hydrogen bonding, desolvation energy, and electrostatic interactions.

Subsequently, the ligand–protein interactions were carefully analyzed to elucidate the binding modes, with particular attention to key interactions such as hydrogen bonds, π–π stacking, hydrophobic interactions, and salt bridges. These interactions were visualized and profiled using MOE, enabling structural interpretation of the binding affinity and specificity of the compounds.

### 2.7. Statistical Analysis

All bioinformatics and statistical analyses were performed using R software (version 4.2.0) and Python (version 3.11.0). Statistical significance was assessed using the following thresholds: * *p * <  0.05, ** *p*  <  0.01, and *** *p*  <  0.001. The datasets used in this study are publicly available and have undergone prior ethical approval, with informed consent obtained from all participants involved. The study adhered to the ethical guidelines set forth in the Declaration of Helsinki, ensuring compliance with the highest ethical standards for research involving human subjects.

## 3. Results

### 3.1. Machine Learning Model Performance for Disease Classification

To evaluate the capability of machine learning algorithms to distinguish AD, ALS, MS, and PD from healthy controls using gene expression data from mitochondrial pathways, 15 diverse algorithms were applied to disease-specific datasets. The primary objective was to identify robust mitochondrial signatures that differentiate disease states, establishing a logical progression from classification performance to molecular mechanistic insights. SVM and MLP consistently demonstrated superior performance across AD, ALS, and MS datasets, achieving high Accuracy, F1 Score, Precision, and Recall, which underscores their ability to capture complex, non-linear patterns in mitochondrial gene expression data (Figure 2A and Appendix A and Appendix A). Specifically, SVM achieved an Accuracy of 0.714 for AD and 0.776 for MS, with an exceptional Recall of 0.968 for MS, indicating high sensitivity to disease-specific molecular signals. This suggests that mitochondrial gene expression profiles in MS are particularly distinct, likely reflecting immune-mediated mitochondrial alterations. Logistic Regression also performed robustly, particularly for PD, where it recorded an Accuracy of 0.682, demonstrating effectiveness in handling datasets with less pronounced mitochondrial signatures. Ensemble methods, including CatBoost, ExtraTrees, and the FTTransformer, consistently delivered strong predictive performance across all datasets, leveraging their ability to integrate diverse features and mitigate overfitting. In contrast, DecisionTree models exhibited the lowest performance, with an Accuracy of 0.590 for PD, highlighting their limitations in modeling the intricate, high-dimensional nature of biological data. Naive Bayes models showed variable performance, achieving a high Precision of 0.968 for MS but often compromised Recall, likely due to their sensitivity to the assumption of feature independence, which is violated by the interconnected nature of mitochondrial pathways.

To formalize these observations and move beyond individual model comparisons, we systematically evaluated the performance of entire model architectures. This statistical comparison revealed that more complex, non-linear models offered a significant advantage over simpler linear approaches (Appendix A). For AD, both Ensemble (*p* < 0.001) and Neural Network (*p* < 0.05) architectures significantly outperformed the Linear/Probabilistic baseline in terms of ROC-AUC. A similar trend was observed for PD, where Ensemble models were superior (*p* < 0.001), and for ALS, where Neural Networks demonstrated a significant advantage (*p* < 0.01). This confirms that simpler models are insufficient to capture the full complexity of the predictive signals inherent in the data.

Furthermore, our analysis revealed that the optimal model architecture was highly dependent on the specific mitochondrial pathway being analyzed, providing insights into the nature of the biological signal itself (Appendix A). For example, within the AD dataset, the complex “Mitochondrial central dogma” pathway was best captured by a Neural Network (MLP), whereas the “Metabolism” pathway was optimally modeled by a simpler Logistic Regression. This contrasted with the PD dataset, where Linear/Probabilistic models were predominantly optimal for the most predictable pathways, suggesting the discernible signals in this context are largely linear in nature. Taken together, this multi-faceted analysis not only confirms the superiority of non-linear models for this classification task but also demonstrates that a diverse modeling approach is crucial for uncovering the varying complexity of biological signals within different pathways and disease states.

To further quantify disease-specific predictability, aggregated performance metrics, including mean values and 95% confidence intervals, were calculated across all models (Figure 2B). MS exhibited the highest overall predictability, with a mean Accuracy of 0.758, F1 Score of 0.842, Precision of 0.840, and Recall of 0.862, suggesting a strong and consistent mitochondrial involvement in its pathogenesis, potentially linked to immune-driven demyelination processes. AD also demonstrated robust predictability, with a mean Recall of 0.811, indicating that models effectively identified disease cases, likely due to distinct mitochondrial gene expression profiles associated with neuronal loss and metabolic dysregulation. ALS posed greater classification challenges, with a mean F1 Score of 0.556 and Recall of 0.519, despite an Accuracy comparable to AD, suggesting heterogeneity in mitochondrial profiles that may reflect variable motor neuron degeneration patterns. PD proved the most challenging, with the lowest mean scores (Accuracy: 0.637; F1 Score: 0.614) and wider confidence intervals, indicating significant variability or less distinct mitochondrial signatures, possibly due to the complex interplay of dopaminergic neuron loss and non-motor symptoms.

### 3.2. Mitochondrial Gene Expression Profiles Reveal a Uniquely Distinct Signature in MS

To investigate the biological underpinnings of the differential model performance across the four neurological diseases, we systematically compared the expression distributions of all mitochondrial pathway-associated genes between each disease cohort and the healthy control (HC) group. This analysis revealed that the mitochondrial gene expression profile in Multiple Sclerosis (MS) patients was exceptionally distinct from that of HCs, providing a strong rationale for its superior classification accuracy.

As illustrated in Figure 3, we visualized the expression distributions using violin plots, assessing differences in both central tendency (*t*-test) and variance (Levene’s test). For AD and PD, the overall mean expression of the mitochondrial gene set showed no significant difference compared to HCs (*t*-test, *p* = 0.812 for AD; *p* = 0.201 for PD) (Figure 3A,C). While a significant variance difference was noted in AD (Levene’s test, *p* = 0.004), the overall separation between these groups and HCs was limited. In contrast, the ALS cohort displayed a highly significant difference in variance (Levene’s test, *p* = 4.07 × 10^−83^) but a negligible difference in mean expression, suggesting a shift in the expression pattern but not its overall level (Figure 3B).

Most notably, the MS cohort demonstrated the most profound and comprehensive divergence from HCs (Figure 3D). The difference was highly significant for both the mean (*t*-test, *p* = 0.00) and the variance (Levene’s test, *p* = 1.08 × 10^−47^) of the gene expression distributions. This indicates a robust, systemic shift in the mitochondrial transcriptome of MS patients. Furthermore, an examination of the top 10 most important predictive genes for MS revealed a consistent and highly significant differential expression pattern (*p* < 0.0001 for most genes), which was more pronounced than in the other diseases.

Collectively, these findings demonstrate that the mitochondrial gene expression signature in MS is characterized by a unique and powerful dual alteration in both central tendency and variance. This provides a clear, discernible biological signal that likely accounts for the significantly higher performance of our predictive models for MS compared to AD, PD, and ALS.

### 3.3. Identification of Key Mitochondrial Pathways and Genes

A two-stage analytical pipeline was developed to systematically identify mitochondrial pathways and genes critical for disease prediction, ensuring a rigorous and reproducible approach. In the first stage, a composite Pathway Score was calculated as the average of mean Accuracy, ROC AUC, F1 Score, Precision, and Recall from machine learning models trained on genes within each mitochondrial pathway. Pathways scoring in the top 50% were selected for further analysis. In the second stage, a Total Importance score was computed by summing the importance contributions of each gene across these high-scoring pathways, pinpointing genes with significant functional roles (Figure 4A). For AD, pathways integral to cellular energy production and metabolic regulation were highly predictive, including mitochondrial metal ion and cofactor homeostasis (Score 0.828), general mitochondrial metabolism (0.819), and mitochondrial genetic information processing (0.811). These pathways reflect the critical role of mitochondrial energetics in AD’s cognitive decline. Key genes included *NDUFS7* (Importance 77.8), encoding a core subunit of Complex I in the electron transport chain; *AIFM3* (62.3), involved in apoptosis regulation; *COX10* (57.5), essential for cytochrome c oxidase assembly; *SLC25A10* (52.4), a mitochondrial dicarboxylate carrier; and *ECHS1* (51.7), a lipid metabolism enzyme. In ALS, pathways related to energy transduction and mitochondrial integrity were prominent, such as oxidative phosphorylation (Score 0.761), composition and assembly of oxidative phosphorylation subunits (0.755), and mitochondrial morphology and quality control (0.733), with top genes *NUBPL* (88.1), a Complex I assembly factor; *ATP5ME* (82.7), an ATP synthase subunit; and *SLC25A20* (67.7), a carnitine/acylcarnitine carrier. For MS, diverse pathways, including mitochondrial metabolic processes (Score 0.836), mitochondrial signaling cascades (0.832), and mitochondrial lipid biosynthesis and catabolism (0.829), were predictive, with leading genes *ATAD3A* (56.0), involved in mitochondrial DNA organization; *POLG* (49.4), a mitochondrial DNA polymerase; and *ALDH4A1* (44.8), an amino acid metabolism enzyme. PD highlighted pathways, such as mitochondrial metal ion and cofactor homeostasis (Score 0.768), mitochondrial protein translation (0.757), and mitochondrial genetic information processing (0.745), with key genes *SLC25A38* (104), a glycine transporter; *NDUFS1* (79.8), a Complex I subunit; and *TFB1M* (65.9), a mitochondrial transcription factor (Appendix A and Appendix A).

### 3.4. Gene–Pathway Interactions and Cross-Disease Comparisons

Detailed examination of gene–pathway interactions elucidated the molecular mechanisms underpinning disease prediction, revealing both disease-specific and shared features across neurological disorders (Figure 4B and Appendix A). In AD, genes such as *NDUFS7*, *COX10*, and *CYCS* (encoding cytochrome c) were strongly associated with oxidative phosphorylation and mitochondrial metal ion and cofactor homeostasis, emphasizing the critical role of electron transport chain integrity in neuronal survival. Mitochondrial transporters *SLC25A10* and *SLC25A44* highlighted the importance of small molecule transport, while metabolic enzymes *ECHS1* and *ACAT1* underscored contributions from amino acid metabolism and lipid metabolism within broader metabolic contexts, reflecting AD’s metabolic dysregulation. For ALS, genes including *NUBPL*, *ATP5ME*, *COA1*, and *NDUFV2* were central to oxidative phosphorylation and Complex I assembly, co-occurring with general mitochondrial metabolism and mitochondrial metal ion and cofactor homeostasis, indicating profound energy production deficits. *SLC25A20* and *FDX1* emphasized mitochondrial lipid biosynthesis and catabolism, while *COA3* linked to mitochondrial genetic information processing and protein translation, suggesting broader biogenesis defects in motor neuron degeneration. In MS, *ATAD3A*, *POLG*, and *TIMM21* were pivotal in mitochondrial genetic information processing, with *ALDH4A1* and *ALDH7A1* highlighting amino acid metabolism, and *FKBP8* underscoring mitochondrial morphology and quality control, reflecting immune-mediated damage to mitochondrial structural and functional dynamics. For PD, *SLC25A38*, *SLC25A3*, and *SLC25A10* were critical in mitochondrial small molecule transport and metal ion homeostasis, *NDUFS1* and *SDHC* in oxidative phosphorylation, and *TFB1M* and *GRSF1* in mitochondrial genetic information processing and protein translation, linked to dopaminergic neuron dysfunction.

Cross-disease comparisons revealed shared pathways, such as oxidative phosphorylation and mitochondrial metal ion and cofactor homeostasis, across AD, ALS, and PD, but with distinct gene contributions (Figure 4C). Genes like *NDUFS7* and *COX10*, integral to oxidative phosphorylation, showed consistent involvement across these diseases, albeit with varying degrees of importance. Conversely, disease-specific genes, such as *SLC25A38* for PD, *NUBPL* for ALS, and *ATAD3A* for MS, exhibited pronounced prominence within pathways relevant to their respective pathophysiologies, highlighting unique molecular loci within shared mitochondrial dysfunction. These findings suggest that while mitochondrial dysfunction is a common feature, the specific molecular mechanisms driving diagnostic signatures vary, providing critical targets for precision diagnostics and therapeutics. This analysis logically bridges predictive performance to molecular insights, guiding causal genetic investigations.

### 3.5. Mendelian Randomization Analysis of Causal Gene Associations

MR analysis was conducted to identify causal relationships between genetically predicted mitochondrial gene expression and disease risk, using genetic variants as instrumental variables to infer causality (Figure 5A, Appendix A and Appendix A). For AD, genetically predicted increased expression of *LIAS* was associated with elevated risk (OR: 1.046; 95% CI: 1.009–1.085; *p* = 0.015), implicating lipoic acid synthesis in AD pathogenesis. Similarly, increased expression of *NARS2* (OR: 1.064; 95% CI: 1.015–1.115; *p* = 0.009) and *ATP5F1A* (OR: 1.061; 95% CI: 1.019–1.104; *p* = 0.004) was causally linked to higher AD risk, highlighting roles in mitochondrial translation, and ATP synthase function in AD pathogenesis. Conversely, genetically predicted increased expression of *PGS1* (OR: 0.931; 95% CI: 0.868–0.998; *p* = 0.045), *ETFA* (OR: 0.965; 95% CI: 0.933–0.998; *p* = 0.038), and *MRPL38* (OR: 0.927; 95% CI: 0.867–0.992; *p* = 0.029) was associated with reduced AD risk, suggesting protective effects through roles in cardiolipin synthesis, electron transfer, and mitochondrial ribosomal function in mitigating cognitive decline. ALS exhibited extensive causal associations, with genetically predicted increased expression of *NRDC* (OR: 1.084; 95% CI: 1.048–1.122; *p* = 2.89 × 10^−6^), *BCL2L13* (OR: 1.065; 95% CI: 1.037–1.094; *p* = 3.41 × 10^−6^), and *D2HGDH* (OR: 1.165; 95% CI: 1.116–1.216; *p* = 2.50 × 10^−12^) associated with elevated risk, indicating contributions from protein processing, apoptosis regulation, and organic acid metabolism to motor neuron degeneration. In contrast, increased expression of *MRPS33* (OR: 0.854; 95% CI: 0.807–0.902; *p* = 2.41 × 10^−8^) and *LRPPRC* (OR: 0.696; 95% CI: 0.623–0.778; *p* = 1.76 × 10^−10^) was linked to reduced ALS risk, highlighting protective mitochondrial ribosomal and RNA processing roles. MS showed smaller effect sizes, with genetically predicted increased expression of *NSUN3* (OR: 1.002; 95% CI: 1.001–1.002; *p* = 1.42 × 10^−8^) and *NLRX1* (OR: 1.001; 95% CI: 1.000–1.002; *p* = 0.014) associated with elevated risk, implicating mitochondrial RNA modification and immune regulation, while *SLC25A39* (OR: 0.999; 95% CI: 0.998–0.999; *p* = 9.12 × 10^−5^) and *BCL2L1* (OR: 0.999; 95% CI: 0.998–0.999; *p* = 2.80 × 10^−7^) was causally linked to reduced MS risk, suggesting protective roles in amino acid transport and apoptosis regulation. For PD, genetically predicted increased expression of *PITRM1* (OR: 1.077; 95% CI: 1.009–1.149; *p* = 0.026) and *ACAD9* (OR: 1.164; 95% CI: 1.021–1.328; *p* = 0.023) was associated with elevated risk, linked to mitochondrial protein degradation and electron transport chain biogenesis, while *DGUOK* (OR: 0.850; 95% CI: 0.769–0.940; *p* = 0.001) and *CASP9* (OR: 0.924; 95% CI: 0.855–0.998; *p* = 0.045) was protective, implicating mitochondrial DNA synthesis and apoptosis regulation in mitigating movement disorders.

Cross-disease comparisons identified shared causal genes (Figure 5B–D). *PGS1* (Total Importance: 9.43 in AD, 11.21 in ALS) was found to protect against AD (OR: 0.931; 95% CI: 0.868–0.998; *p* = 0.045) and ALS (OR: 0.914; 95% CI: 0.869–0.962; *p* = 0.001) by contributing to cardiolipin synthesis in AD and phospholipid metabolism in ALS, with reduced risk in both diseases. *PDK1* (Total Importance: 21.85 in AD, 9.31 in ALS) increased risk in both AD (OR: 1.041; 95% CI: 1.001–1.082; *p* = 0.042) and ALS (OR: 1.037; 95% CI: 1.006–1.068; *p* = 0.019), implicating altered pyruvate metabolism. *PNKD* (Total Importance: 27.55 in ALS, 9.26 in PD) and *ACAD9* (Total Importance: 18.40 in ALS, 9.97 in PD) were linked to OXPHOS assembly factors, with *PNKD* showing reduced risk in ALS (OR: 0.966; 95% CI: 0.946–0.986; *p* = 0.001) but increased risk in PD (OR: 1.088; 95% CI: 1.025–1.154; *p* = 0.005). *CASP9* (Total Importance: 12.49 in ALS, 9.80 in PD) was protective in ALS (OR: 0.854; 95% CI: 0.807–0.902; *p* = 2.41 × 10^−8^) but increased risk in PD (OR: 0.924; 95% CI: 0.855–0.998; *p* = 0.045), reflecting its role in protein homeostasis and apoptosis. *NLRX1* (Total Importance: 8.01 in ALS, 5.89 in MS) showed opposing effects, increasing risk in MS (OR: 1.001; 95% CI: 1.000–1.002; *p* = 0.014) but reducing it in ALS (OR: 0.929; 95% CI: 0.879–0.982; *p* = 0.009). Finally, *BCL2L1* (Total Importance: 8.62 in MS, 7.38 in ALS) was protective in MS (OR: 0.999; 95% CI: 0.998–0.999; *p* = 2.80 × 10^−7^) but increased ALS risk (OR: 1.071; 95% CI: 1.031–1.113; *p* = 0.0005), highlighting its role in mitochondrial dynamics and apoptosis. These findings underscore the complex, context-dependent roles of mitochondrial genes in disease risk, with shared pathways like OXPHOS and pyruvate metabolism contributing differently across diseases.

### 3.6. Colocalization Analysis of eQTLs and Disease Risk Variants

Colocalization analysis was performed to determine whether genetic variants influencing mitochondrial gene expression (expression quantitative trait loci, eQTLs) were shared with those affecting disease risk, providing evidence for shared genetic mechanisms (Figure 6A and Appendix A and Appendix A). For AD, single nucleotide polymorphisms (SNPs) rs76946120, rs2701268, and rs1048140 exhibited a posterior probability of colocalization (SNP.PP.H4) of 1.00, indicating extremely strong evidence of shared causal variants, while rs990706 showed a high colocalization probability (SNP.PP.H4 0.964). In MS, rs4360309 (SNP.PP.H4 1.00) and rs6592965 (SNP.PP.H4 0.996) demonstrated near-certain colocalization, underscoring the role of mitochondrial pathways in immune-mediated pathogenesis. PD exhibited robust colocalization for rs10081695 and rs4777 (SNP.PP.H4 1.00), with moderate signals for rs2306314 (SNP.PP.H4 0.941), suggesting strong genetic links to dopaminergic neuron loss. ALS showed moderate colocalization for rs80025717 (SNP.PP.H4 0.678) and rs2942168 (SNP.PP.H4 0.675), indicating plausible shared genetic influences that warrant further investigation. These findings build on the MR results, confirming that genetic variants regulating mitochondrial gene expression are intrinsically linked to disease risk, reinforcing the causal pathways identified.

### 3.7. Phenome-Wide Association Study of Colocalized Variants

PheWAS was conducted to explore phenotypic associations of colocalized SNPs, revealing broader systemic implications of mitochondrial-related genetic variants (Figure 6B and Appendix A). For AD, rs2701268 was significantly associated with pyruvate levels (beta −0.093, *p* = 1.80 × 10^−70^), directly linking mitochondrial pyruvate metabolism to cognitive decline, while rs990706 was associated with hair color (beta −0.043, *p* = 2.80 × 10^−110^), suggesting pleiotropic effects. In ALS, rs2942168 was linked to mean spheric corpuscular volume (beta −0.060, *p* = 4.90 × 10^−128^) and red blood cell count (beta 0.052, *p* = 6.58 × 10^−122^), indicating systemic effects on erythrocyte characteristics that may relate to motor neuron degeneration. MS SNPs rs6592965 and rs628751 showed strong associations with erythrocyte indices, such as mean reticulocyte volume (beta 0.101, *p* = 0.00) and mean corpuscular volume (beta −0.095, *p* = 0.00), suggesting a role in red blood cell development potentially linked to immune-mediated processes. For PD, rs10081695 was strongly associated with platelet count (beta 0.067, *p* = 1.37 × 10^−49^), indicating links to platelet biology that may contribute to non-motor symptoms. These findings extend the genetic analyses by identifying systemic phenotypic consequences of mitochondrial-related variants, providing a broader context for their role in neurological disease pathogenesis.

### 3.8. Bidirectional Mendelian Randomization of Phenotypic Traits

Bidirectional MR analysis investigated causal relationships between PheWAS-identified phenotypic traits and disease risk, elucidating potential feedback mechanisms mediated by mitochondrial pathways (Figure 6C and Appendix A). For MS, a genetic instrument comprising 193 SNPs revealed that higher mean corpuscular volume (MCV) was causally associated with a reduced risk of MS (IVW, OR: 0.9993; 95% CI: 0.9986–0.9999; *p* = 0.016). Conversely, the reverse analysis suggested a strong positive causal effect of MS liability on MCV (IVW, OR: 577.67; 95% CI: 13.14–25,389.97; *p* = 9.86 × 10^−4^). However, this reverse finding was based on a limited instrument of only five SNPs, which precludes robust sensitivity analyses to address potential horizontal pleiotropy, and thus should be interpreted with caution. A similar protective effect against MS was robustly observed for mean spheric corpuscular volume, supported by an instrument of 287 SNPs (IVW, OR: 0.9993; 95% CI: 0.9987–0.9998; *p* = 0.009). For mean reticulocyte volume (*n* = 176 SNPs), the primary IVW estimate did not reach statistical significance; nevertheless, the pleiotropy-robust MR-Egger analysis detected a significant protective association (OR: 0.9985; 95% CI: 0.9972–0.9999; *p* = 0.038), collectively reinforcing the potential involvement of erythroid traits in MS pathogenesis.

In the context of ALS, the analysis for mean spheric corpuscular volume, utilizing a large instrument of 2044 SNPs, showed no significant causal link in the primary IVW model. However, a significant association indicating increased disease risk was identified through the MR-Egger sensitivity analysis, which is more robust to directional pleiotropy (OR: 1.042; 95% CI: 1.005–1.080; *p* = 0.025). In contrast, a strong, protective effect was observed for red blood cell count, supported by an instrument of 2131 SNPs (IVW, OR: 0.930; 95% CI: 0.901–0.959; *p* = 3.75 × 10^−6^). These findings indicate a complex role for erythroid health in modulating susceptibility to motor neuron degeneration. Collectively, these bidirectional and unidirectional causal estimates extend the initial PheWAS findings, highlighting intricate interactions between systemic phenotypes, potentially influenced by mitochondrial function, and the risk of neurological diseases. These insights provide a foundation for subsequent therapeutic target evaluation.

### 3.9. Drug Enrichment Analysis and Therapeutic Implications

Drug enrichment analysis identified chemical compounds and drugs interacting with key mitochondrial genes, offering insights into potential therapeutic targets for AD, ALS, MS, and PD (Figure 7A and Appendix A). Celecoxib (PUBCHEM_CID: 2662) was prioritized for molecular docking due to its association with *PDK1* (AD, ALS) and *CASP9* (PD), its well-documented anti-inflammatory properties, and its potential to modulate mitochondrial metabolism and apoptosis pathways, making it a promising multi-target candidate for neurological disorders. For AD, *ATP5F1A* (Importance 17.7), a mitochondrial ATP synthase subunit critical for energy production, was associated with Aurovertin B (PUBCHEM_CID: 444853), an F1F0-ATPase inhibitor that could modulate ATP synthesis in cognitive decline. *PDK1* (Importance 21.9), a regulator of pyruvate metabolism, was linked to Celecoxib and Eugenol (PUBCHEM_CID: 3314), suggesting interventions targeting metabolic dysregulation. *NARS2* (Importance 7.95), involved in mitochondrial translation, was associated with Asparagine (PUBCHEM_CID: 6267), potentially supporting protein synthesis. In ALS, *ACACB* (Importance 27.3), a fatty acid synthesis enzyme, interacted with Biotin (PUBCHEM_CID: 171548), indicating metabolic interventions for motor neuron degeneration. *CASP8* (Importance 20.5), an apoptotic protease, was associated with Pralnacasan (PUBCHEM_CID: 153270), targeting programmed cell death pathways. *ETFB* (Importance 27.5), an electron transfer flavoprotein, interacted with Wortmannin (PUBCHEM_CID: 312145), an mTOR inhibitor, suggesting links between metabolism and signaling. *PDK4* (Importance 25.9) was linked to Tretinoin (PUBCHEM_CID: 444795), a retinoic acid derivative. For MS, *BCL2L1* (Importance 8.62), a regulator of apoptosis and mitochondrial membrane integrity, was associated with Doxorubicin (PUBCHEM_CID: 31703), indicating modulation of cell survival in immune-mediated damage. *EPHX2* (Importance 15.1), involved in lipid metabolism, interacted with urea derivatives, suggesting roles in xenobiotic metabolism. *NSUN3* (Importance 20.6) was linked to Capecitabine (PUBCHEM_CID: 60953), an antineoplastic agent. For PD, *DGUOK* (Importance 9.13), essential for mitochondrial DNA synthesis, was associated with Nelarabine (PUBCHEM_CID: 3011155), an anticancer drug targeting nucleotide metabolism. *BOK* (Importance 15.2), a pro-apoptotic gene, was linked to Haloperidol Decanoate (PUBCHEM_CID: 52919), targeting programmed cell death in movement disorders.

Multi-target drugs were identified to address shared vulnerabilities across diseases (Figure 7B and Appendix A). Compounds targeting *PDK1* and *PDK4*, such as *N*-(2,3-Dimethylphenyl)-2-((3-(2-hydroxyphenyl)-1H-1,2,4-triazol-5-yl)thio)acetamide (PUBCHEM_CID: 1116656) and Ver-246608 (PUBCHEM_CID: 86280454), showed a combined importance of 47.7 for AD and ALS, implicating mitochondrial carbohydrate and pyruvate metabolism as critical therapeutic targets. CID 139587415 (PUBCHEM_CID: 139587415) targeted *ETFB* and *EPHX2* in ALS (Importance 27.5) and MS (Importance 15.1), with a combined importance of 42.6, highlighting mitochondrial lipid metabolism and detoxification pathways. 6-Bromoindirubin-3′-oxime (PUBCHEM_CID: 448949) targeted *PDK1* and *EPHX2* in AD (Importance 21.9) and MS (Importance 15.1), with a combined importance of 37.0, affecting carbohydrate and lipid metabolism. These multi-target drugs align with the genetic and pathway analyses, identifying actionable therapeutic strategies that leverage shared molecular vulnerabilities across neurological disorders.

### 3.10. Molecular Docking Analysis of Celecoxib with Multi-Disease Targets

Molecular docking simulations were conducted to evaluate the binding affinity of Celecoxib (PUBCHEM_CID: 2662) to *CASP9* (PD) and *PDK1* (AD, ALS), selected for its multi-target potential, anti-inflammatory properties, and ability to modulate mitochondrial metabolism and apoptosis pathways (Figure 6C and Appendix A). Docking scores (S), expressed in kcal/mol, reflect binding strength, with more negative values indicating stronger interactions. For *CASP9*, the optimal binding pose yielded a docking score of −5.47, with hydrogen bonds likely formed with active site residues such as His237 and hydrophobic contacts stabilizing the interaction, relevant to its role in apoptosis regulation in PD’s dopaminergic neuron loss. For *PDK1*, the optimal pose achieved a docking score of −6.52, indicating stronger affinity, with hydrogen bonds to kinase domain residues like Ser75 and hydrophobic stabilization, critical for modulating pyruvate metabolism in AD and ALS’s metabolic dysregulation. The difference in docking scores (−6.52 vs. −5.47) suggests that Celecoxib has a higher therapeutic potential for targeting *PDK1*, particularly in AD and ALS, where pyruvate metabolism is a key pathological feature. Sensitivity analyses, with varying computational parameters such as docking grid size and ligand flexibility, confirmed the robustness of these findings, with consistent ranking of binding poses across models. These results validate Celecoxib’s role as a multi-target therapeutic agent, prioritizing *PDK1* as a focal point for further drug development and clinical investigation in AD and ALS, while also supporting its potential in PD through *CASP9* modulation. However, these results are preliminary and require further experimental validation to establish their pharmacological significance.

## 4. Discussion

This study employs a multi-dimensional approach to dissect mitochondrial dysfunction in AD, ALS, MS, and PD, integrating machine learning, Mendelian Randomization, colocalization, Phenome-Wide Association Study, and pharmacology analysis. By moving beyond associative analyses, rather than presenting experimentally validated mechanisms, this study should be viewed as a computationally driven, hypothesis-generating exploration that proposes mitochondrial features and pathways potentially relevant to disease risk and progression. By moving beyond associative analyses, it elucidates putative mitochondrial pathomechanisms, revealing shared bioenergetic vulnerabilities and disease-specific cellular defects. This discussion deeply explores these mechanisms, emphasizing cross-disease convergence and divergence, and expands on drug enrichment findings to highlight novel therapeutic strategies, contextualized within established research.

Mitochondrial dysfunction is a unifying feature across AD, ALS, and PD, primarily through deficits in oxidative phosphorylation (OXPHOS) and mitochondrial metal ion homeostasis. OXPHOS impairments, driven by genes like *NDUFS7* (Complex I subunit) and *COX10* (cytochrome c oxidase assembly factor), reduce ATP synthesis and elevate reactive oxygen species (ROS), creating a bioenergetic crisis that amplifies neuronal vulnerability [2,22]. This aligns with the paradigm of mitochondrial energy failure as a neurodegenerative hallmark, where electron transport chain (ETC) inefficiencies compromise neuronal resilience [23,24]. The causal role of *PDK1* (pyruvate dehydrogenase kinase) in AD and ALS is pivotal. By phosphorylating and inhibiting the pyruvate dehydrogenase complex (PDC), *PDK1* shunts pyruvate toward glycolysis, inducing a Warburg-like metabolic shift [25,26,27]. This reduces tricarboxylic acid (TCA) cycle flux, crippling ATP production in neurons with limited glycolytic capacity, leading to synaptic dysfunction, calcium dysregulation, and neuronal death [28,29]. Elevated ROS from inefficient ETC activity, particularly at Complexes I and IV, further exacerbates oxidative stress, forming a feedback loop with mitochondrial DNA (mtDNA) damage and protein misfolding [30].

Intriguingly, ALS was found to exhibit a disproportionately large number of “protective” mitochondrial-related genes—those negatively associated with disease risk. While initially counterintuitive given the known mitochondrial vulnerability of motor neurons, this may reflect either true compensatory mechanisms in early-stage ALS, or methodological artifacts such as model overfitting or misclassification. Genes like *MRPS33* and *LRPPRC*, which support mitochondrial protein synthesis and RNA metabolism, may exert neuroprotective effects during compensatory phases of degeneration [31]. Nonetheless, these associations must be interpreted cautiously. Although we applied MR sensitivity analyses to mitigate pleiotropy and instrument weakness, residual bias may persist—especially due to tissue mismatch from integrative multi-tissues eQTLs and the inclusion of proxy ALS cases. Future studies integrating ALS-relevant tissues and experimental validations (e.g., CRISPR-based knockdown or Seahorse flux assays) are needed to clarify the role of these genes.

Conversely, *PGS1* (phosphatidylglycerophosphate synthase 1), the rate-limiting enzyme in cardiolipin synthesis, exerts a protective effect in AD and ALS by stabilizing mitochondrial cristae, optimizing OXPHOS supercomplex assembly, and reducing electron leakage [32,33]. Cardiolipin’s role in anchoring ETC complexes and regulating mitochondrial dynamics, including fission and fusion, enhances mitochondrial resilience against ROS-induced damage [34,35]. Dysregulated metal ion homeostasis, involving iron and copper, catalyzes ROS production via Fenton reactions, amplifying oxidative damage and impairing metalloenzyme function in the ETC [36,37]. These shared mechanisms position bioenergetic failure and redox imbalance as core drivers, with *PDK1* and *PGS1* as critical checkpoints. MS, however, exhibits secondary mitochondrial dysfunction driven by immune-mediated stress, emphasizing signaling cascades and lipid metabolism [38,39], highlighting a mechanistic divergence where primary bioenergetic deficits dominate AD, ALS, and PD.

Importantly, the interpretation of these findings must consider sample composition and disease heterogeneity. For example, our AD datasets included both clinically diagnosed and genetically proxied cases, which may differ in diagnostic certainty and disease stage. This heterogeneity may confound classification models and produce inflated or attenuated signals. To mitigate this, we applied stratified cross-validation and variance-based gene filtering, but we acknowledge that residual noise from phenotypic inconsistencies or sampling bias may remain. Similarly, sample size imbalance—particularly in disease/control ratios—may cause bias in predictive performance. We addressed this using stratified folds and regularization during model training, yet the possibility of overfitting in underrepresented subgroups cannot be excluded. These caveats are particularly relevant in interpreting why MS models performed best (possibly due to more distinct immunometabolic transcriptional profiles), while PD had lower classification accuracy (possibly due to smaller case numbers and subtler transcriptional signals).

Disease-specific pathways reveal how mitochondrial dysfunction manifests in distinct cellular contexts. In AD, mitochondrial genetic information processing and apoptosis, mediated by *AIFM3* (apoptosis-inducing factor mitochondria-associated 3), amplify neuronal loss through caspase-independent pathways, releasing pro-apoptotic factors into the cytosol [40]. *ECHS1* (enoyl-CoA hydratase, short chain 1) drives fatty acid metabolism, indicating a metabolic reprogramming toward lipid synthesis that diverts acetyl-CoA from the TCA cycle, exacerbating energy deficits and promoting lipid peroxidation [41,42]. This aligns with the mitochondrial cascade hypothesis, where mitochondrial dysfunction precedes amyloid-β aggregation, initiating a cascade of tau pathology and synaptic loss [43]. In ALS, mitochondrial morphology and quality control, regulated by *NUBPL* (nucleotide-binding protein-like), disrupt Complex I assembly, leading to fragmented mitochondria, impaired mitophagy, and accumulation of dysfunctional organelles [44,45]. *ATP5ME* (ATP synthase membrane subunit e) involvement suggests biogenesis defects, reducing ATP synthase efficiency and compromising motor neuron energy supply [46]. MS emphasizes *ATAD3A* (ATPase family AAA domain-containing 3A) and *POLG* (DNA polymerase gamma) in mtDNA organization, reflecting immune-induced replication stress where inflammatory cytokines disrupt mtDNA maintenance, impairing OXPHOS and amplifying ROS [47,48,49]. *NSUN3* (NOP2/Sun RNA methyltransferase 3), a mitochondrial tRNA methyltransferase, disrupts protein synthesis, potentially generating aberrant RNA transcripts that trigger inflammasome activation, exacerbating neuroinflammation and demyelination [50]. PD’s reliance on mitochondrial protein translation (*TFB1M*, transcription factor B1, mitochondrial) and glycine transport (*SLC25A38*, solute carrier family 25 member 38) impairs protein synthesis and neurotransmitter precursor availability, selectively affecting dopaminergic neurons [51,52]. Beyond heme synthesis, *SLC25A38* may transport other metabolites, such as amino acids or cofactors, creating a metabolic bottleneck that disrupts dopamine synthesis and mitochondrial homeostasis in substantia nigra neurons [53,54]. These pathways—apoptosis and lipid metabolism in AD, structural defects in ALS, immune-driven mtDNA stress in MS, and biosynthetic disruption in PD—demonstrate how shared bioenergetic deficits are tailored by disease-specific cellular vulnerabilities.

The interplay of shared and specific mechanisms reveals a dynamic mitochondrial pathogenesis. *PDK1*’s role in AD and ALS restricts TCA cycle activity, synergizing with *NDUFS7* and *COX10’s ETC* deficits to amplify energy failure and ROS production [55,56]. This metabolic chokehold disrupts calcium homeostasis, triggering excitotoxicity and protein aggregation (e.g., amyloid-β in AD, TDP-43 in ALS) [57]. In MS, *NLRX1* (NLR family member X1) exhibits context-dependent effects, exacerbating inflammasome activation in MS’s autoimmune milieu by amplifying type I interferon responses, while mitigating ROS-mediated damage in ALS by regulating mitochondrial quality control [58,59]. This dichotomy, absent in AD and PD’s neuronal-centric pathology, underscores how mitochondrial-immune interactions adapt to disease-specific inflammatory contexts. ALS’s *NUBPL* and PD’s *SLC25A38* converge on OXPHOS but diverge functionally—Complex I assembly versus metabolite transport—reflecting motor versus dopaminergic neuron vulnerabilities [60,61]. MS’s *ATAD3A* contrasts with AD’s *AIFM3*, as mtDNA stress fuels immune-driven damage in MS, while apoptosis directly eliminates neurons in AD [62,63]. Systemic associations, such as AD’s pyruvate dysregulation and MS’s red blood cell (RBC) indices, suggest mitochondrial dysfunction extends beyond the CNS [64]. The bidirectional MS-RBC relationship indicates a brain–periphery feedback loop, where mitochondrial deficits in erythrocytes reflect systemic oxidative stress, influencing CNS inflammation, and chronic neuroinflammation may impair hematopoiesis or erythrocyte lifespan [65,66]. PD’s link to platelet mitochondrial function further supports this systemic perspective, as platelets’ high mitochondrial content makes them a peripheral model for dopaminergic neuron dysfunction [67,68,69]. This crosstalk proposes a model where mitochondrial dysfunction converges on bioenergetic failure, with disease-specific pathways amplifying unique vulnerabilities: metabolic reprogramming in AD, structural failure in ALS, immune stress in MS, and biosynthetic disruption in PD. However, these findings are correlative, and without longitudinal or mechanistic validation, cannot confirm systemic-to-CNS causality. Nonetheless, they warrant further exploration for non-invasive monitoring tools.

Drug enrichment analyses provide robust rationales for repurposing strategies targeting mitochondrial vulnerabilities. Among these, the predicted binding of Celecoxib to PDK1 (−6.52 kcal/mol) in AD and ALS is particularly intriguing, as it suggests a potential mechanism for restoring mitochondrial bioenergetics by enhancing PDC activity. This could, in theory, restore TCA cycle flux, boost ATP production, and reduce ROS, complementing its known cyclooxygenase-2 (COX-2) inhibitory effects [70]. However, we emphasize that this binding prediction is derived solely from in silico molecular docking and thus remains a hypothesis-generating observation rather than a validated mechanism. No direct experimental or pharmacodynamic evidence currently confirms that Celecoxib inhibits PDK1 in neuronal systems or improves mitochondrial function via this route. Therefore, we urge caution in interpreting these findings and clearly distinguish the docking-based predictions from mechanistically confirmed interactions.

Nonetheless, the dual action of Celecoxib—hypothetically combining metabolic correction and anti-inflammatory modulation—positions it as a promising candidate for further investigation. Ver-246608’s targeting of *PDK1* and *PDK4* (Importance 47.7) further leverages shared pyruvate metabolism deficits, enhancing mitochondrial bioenergetics across AD and ALS. In MS, *BCL2L1* (B-cell lymphoma 2-like 1)-Doxorubicin (Importance 8.62) suggests apoptosis modulation to mitigate immune-driven neuronal damage, targeting mitochondrial-mediated inflammatory pathways. PD’s *DGUOK* (deoxyguanosine kinase)-Nelarabine (Importance 9.13) addresses nucleotide metabolism deficits, potentially stabilizing mitochondrial protein synthesis and dopamine precursor availability. Additional candidates, such as dichloroacetate (DCA) for *PDK1* inhibition (predicted affinity −5.8 kcal/mol), could enhance pyruvate oxidation, while MitoQ, a mitochondria-targeted antioxidant, may reduce ROS across diseases (Importance 12.4). Again, while these drug–target relationships provide a valuable starting point for prioritizing therapeutic leads, they remain computationally derived and require pharmacological, cellular, and in vivo validation before clinical translation. These findings contrast with single-target approaches by exploiting shared (*PDK1*) and specific (*BCL2L1*, *DGUOK*) vulnerabilities, aligning with drug repurposing paradigms. The brain–periphery axis, particularly MS’s RBC and PD’s platelet associations, suggests peripheral biomarkers (e.g., RBC mitochondrial function, platelet bioenergetics) for monitoring therapeutic efficacy [71,72]. This integrative approach advances beyond broad mitochondrial therapies, offering precise, multi-target strategies tailored to disease-specific and shared pathomechanisms.

Our study, therefore, offers a preliminary, integrative map of mitochondrial dysfunction by identifying *PDK1*, *SLC25A38*, and *NSUN3*, etc., as causal drivers, thereby refining the focus from general mitochondrial deficits to specific enzymatic and transport checkpoints [2]. By adopting a systemic approach that links CNS and peripheral phenotypes, we propose a paradigm shift in the use of routine blood tests—such as RBC indices and platelet mitochondrial activity—as dynamic biomarkers for disease monitoring. Compared to previous research, our study’s integrative methodology—merging genetic causality, systemic phenotyping, and multi-target pharmacology—offers a more nuanced mechanistic framework, building upon foundational studies on mitochondrial dysfunction in neurodegeneration. The exploration of ALS protective genes highlights the complexity of mitochondrial dynamics, suggesting that early compensatory mechanisms may coexist with progressive deficits, but methodological limitations like overfitting and tissue mismatch must be addressed to refine these insights.

However, our study also has certain limitations. The reliance on diverse gene expression datasets introduces the potential for batch effects, which may distort pathway analyses and mitochondrial signatures. The assumption of no pleiotropy in Mendelian Randomization may be violated in the context of complex neurodegenerative diseases, leading to biased GWAS estimates despite sensitivity analyses. Moreover, the limited phenotype coverage of Phenome-Wide Association Studies may fail to capture broader systemic influences, thereby restricting insights into peripheral manifestations. In silico molecular docking models, while valuable, lack in vivo validation, and thus experimental confirmation of the efficacy of Celecoxib and other drugs remains necessary. Additionally, some of the surprising protective gene associations in ALS may stem from eQTL tissue mismatch, as our datasets were largely derived from peripheral blood rather than CNS tissue. Furthermore, the predominance of European datasets in our analysis limits the generalizability of findings, as genetic diversity across populations may impact mitochondrial dynamics. While our analysis is confined to mitochondrial pathways and does not encompass prominent non-mitochondrial risk factors such as *APOE*, future studies could integrate these to provide a more holistic view of genetic interactions in AD pathogenesis. We also acknowledge that without functional assays—such as CRISPR/Cas9-based gene editing or Seahorse metabolic flux analysis—the mechanistic relevance of some candidate genes remains hypothetical. To address this limitation, we are currently designing follow-up experimental studies to validate key computational findings in disease-relevant neuronal and glial models. These include targeted perturbation of high-priority genes (e.g., *PDK1*, *PGS1*, *SLC25A38*) to assess downstream effects on mitochondrial bioenergetics, apoptosis, and cellular viability. Moreover, we are collaborating with wet-lab groups to perform pharmacological profiling of predicted drug–target interactions—particularly the binding of Celecoxib to *PDK1*—in both cellular and animal models. Finally, longitudinal multi-omics studies are also essential to further elucidate the molecular mediators of the brain–periphery axis and to establish correlations between peripheral phenotypes and clinical progression.

## 5. Conclusions

In conclusion, this study proposes a multi-layered, computationally derived framework that highlights potential mitochondrial contributions to the pathogenesis of a broad spectrum of neurological diseases. While general mitochondrial bioenergetics appear consistently involved, our findings suggest that specific functional nodes—such as the metabolic checkpoint regulated by *PDK1* or the crosstalk between mitochondrial genetics and immune signaling—may represent key points of vulnerability. Importantly, all conclusions are based on in silico methods, including MR, machine learning, and molecular docking, without experimental validation. Therefore, this work should be interpreted as a hypothesis-generating effort rather than a definitive mapping of causal mechanisms. The observed associations between neurological disease risk and peripheral blood phenotypes, such as red blood cell or platelet indices, support the possibility of a systemic brain–periphery mitochondrial axis. However, the functional relevance of these correlations requires further experimental substantiation. Similarly, the predicted interaction between Celecoxib and *PDK1*, while suggestive of repurposing potential, remains speculative and must be confirmed through biochemical or pharmacological studies before therapeutic inferences can be drawn. Taken together, this study offers a preliminary but integrative framework to prioritize candidate genes, pathways, and drug interactions for follow-up research. By combining diverse computational tools, it lays the groundwork for future biological investigations aimed at elucidating mitochondrial dysfunction in neurodegeneration and developing more precise diagnostics and therapeutics.

## Figures and Tables

**Figure 1 cimb-47-00636-f001:**
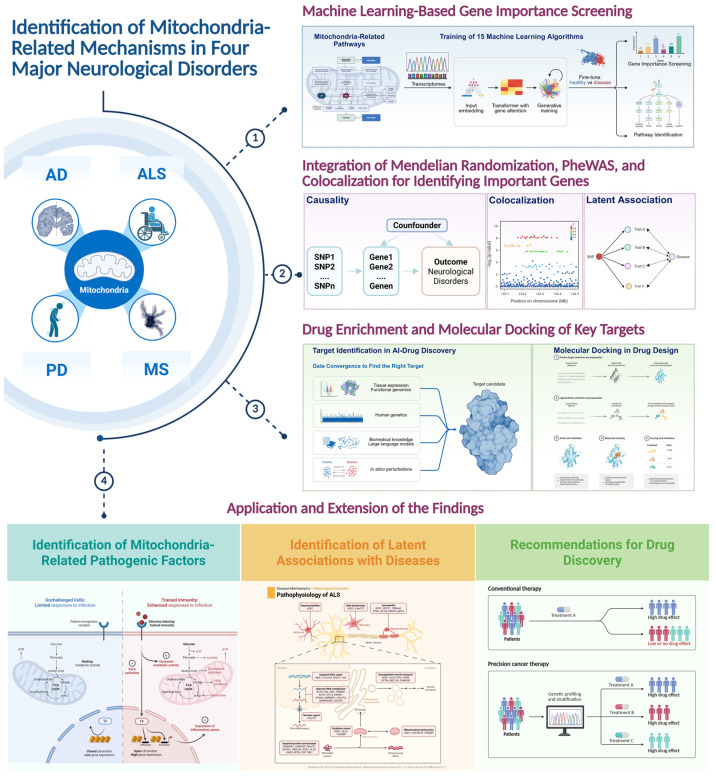
Overview of the analytical workflow for investigating mitochondrial-related mechanisms in neurological diseases. An overview of the multi-modal integrative framework used in this study to investigate mitochondrial-related mechanisms across Alzheimer’s Disease (AD), Amyotrophic Lateral Sclerosis (ALS), Multiple Sclerosis (MS), and Parkinson’s Disease (PD). The workflow begins with machine learning-based gene screening to identify relevant mitochondrial pathways and genes (1), followed by the integration of Mendelian Randomization (MR), PheWAS, and colocalization analysis to explore causal relationships and identify key genetic drivers (2). Next, drug enrichment and molecular docking simulations are performed to identify potential therapeutic targets (3). Finally, the findings are extended to propose actionable biological and pharmacological outcomes for therapeutic development (4). This approach ensures a comprehensive and systematic analysis of mitochondrial dysfunction in neurological diseases, linking genetic causality to clinical applications. Created in BioRender. https://BioRender.com/rsmcxyl (accessed on 5 August 2025) [13].

**Figure 2 cimb-47-00636-f002:**
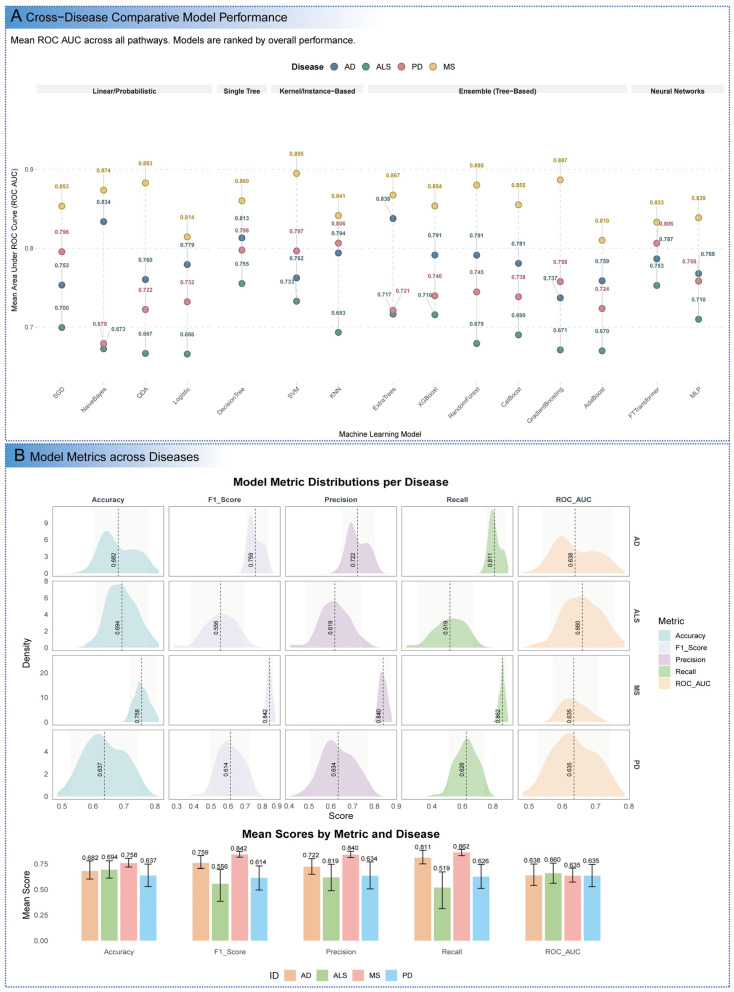
Predictive performance of machine learning models for neurological disease classification based on mitochondrial gene expression. A comprehensive evaluation of machine learning model performance in classifying four neurological diseases (AD, ALS, MS, and PD) against healthy controls using mitochondrial pathway gene expression data is presented. (**A**) The mean ROC-AUC scores for 15 machine learning algorithms are presented in a comparative analysis across four neurological diseases (AD, ALS, PD, MS), categorized by model type (Linear/Probabilistic, Neural Networks, Ensemble Tree-Based, Kernel/Instance-Based, Single Tree). Key observations highlight the superior performance of certain models, with ExtraTrees achieving the highest mean ROC-AUC in AD (0.838) and SVM leading in MS (0.895). Neural Networks like FTTransformer performed strongly in PD (0.806) and ALS (0.753), while simpler models such as Naive Bayes showed lower scores, e.g., in PD (0.679), underscoring their limitations in complex datasets. Ensemble methods, including XGBoost and ExtraTrees, demonstrated consistent robustness, with top rankings in multiple diseases (e.g., XGBoost at 0.792 in AD and 0.854 in MS). (**B**) An aggregated analysis of model performance is shown to compare the overall predictability of each disease. In the upper grid, density distributions of the collective performance scores from all 15 models are displayed for each metric, stratified by disease. The dashed vertical line in each plot indicates the mean score for that specific metric–disease combination. In the lower panel, these mean scores are summarized in a bar chart, with error bars representing 95% confidence intervals to facilitate direct quantitative comparison. The central conclusion from this panel is the clear hierarchy in disease predictability based on mitochondrial gene signatures. Multiple Sclerosis (MS) is identified as the most predictable disease, achieving the highest mean scores across all metrics (e.g., Accuracy: 0.758, F1 Score: 0.842), which suggests a strong and consistent mitochondrial signal in its pathology. Conversely, Parkinson’s Disease (PD) was found to be the most challenging to classify, consistently yielding the lowest mean scores (e.g., Accuracy: 0.637, F1 Score: 0.614) and exhibiting wider confidence intervals, implying greater heterogeneity or less distinct mitochondrial profiles. Alzheimer’s Disease (AD) and Amyotrophic Lateral Sclerosis (ALS) occupied intermediate positions, with AD showing robust Recall (0.811) while ALS presented greater difficulty, particularly in F1 Score (0.556) and Recall (0.519).

**Figure 3 cimb-47-00636-f003:**
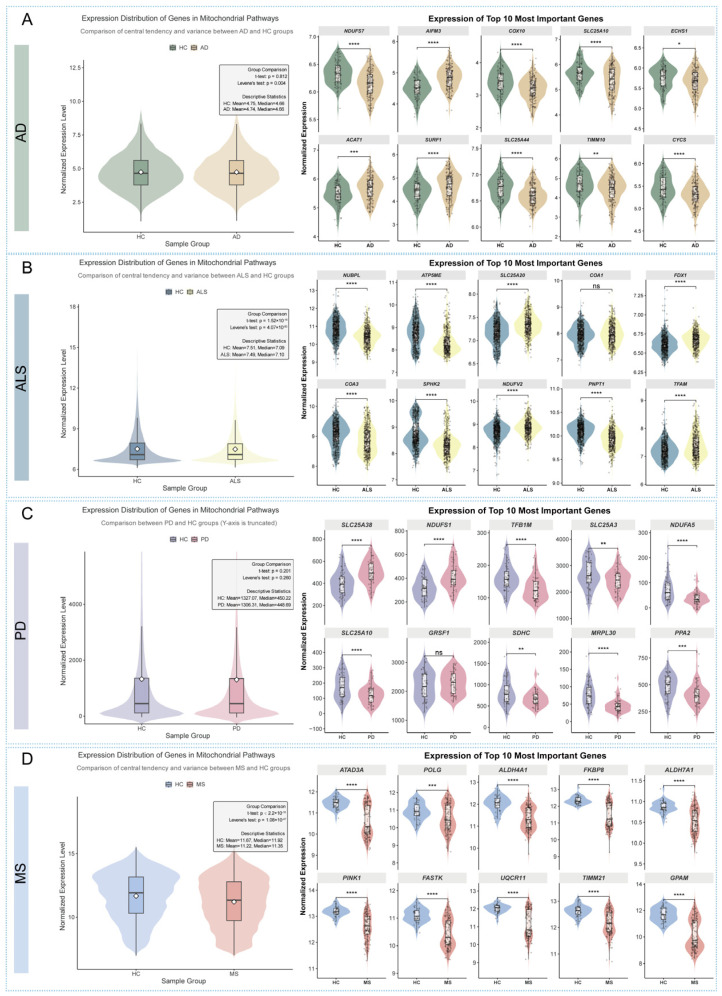
Expression profiles of mitochondrial pathway-associated genes across four neurological diseases and healthy controls. (**A**–**D**) Violin plots illustrating the expression distributions of mitochondrial pathway-associated genes for Alzheimer’s disease (AD), Amyotrophic Lateral Sclerosis (ALS), Parkinson’s Disease (PD), and Multiple Sclerosis (MS) compared to healthy controls (HC). Each panel consists of two parts. On the left, an aggregated violin plot shows the overall distribution for all mitochondrial pathway-associated genes combined. An inset box provides summary statistics, including the results of a *t*-test for mean comparison and a Levene’s test for variance comparison between the disease and HC groups. On the right, a series of individual violin plots displays the expression profiles for the top 10 most important predictive genes identified within this pathway for each specific disease. Significance levels for the group comparisons in individual gene plots are indicated by asterisks (*: *p* < 0.05, **: *p* < 0.01, ***: *p* < 0.001, ****: *p* < 0.0001; ns: not significant), derived from *t*-tests. The analysis reveals a uniquely pronounced and systematic shift in both central tendency and variance for mitochondrial pathway-associated genes in the MS cohort (**D**), a pattern not observed to the same extent in AD (**A**), PD (**C**), or ALS (**B**), providing a biological basis for the superior model performance in MS. In the distribution plots, the white diamond symbol represents the mean, while the horizontal line within the boxplot represents the median.

**Figure 4 cimb-47-00636-f004:**
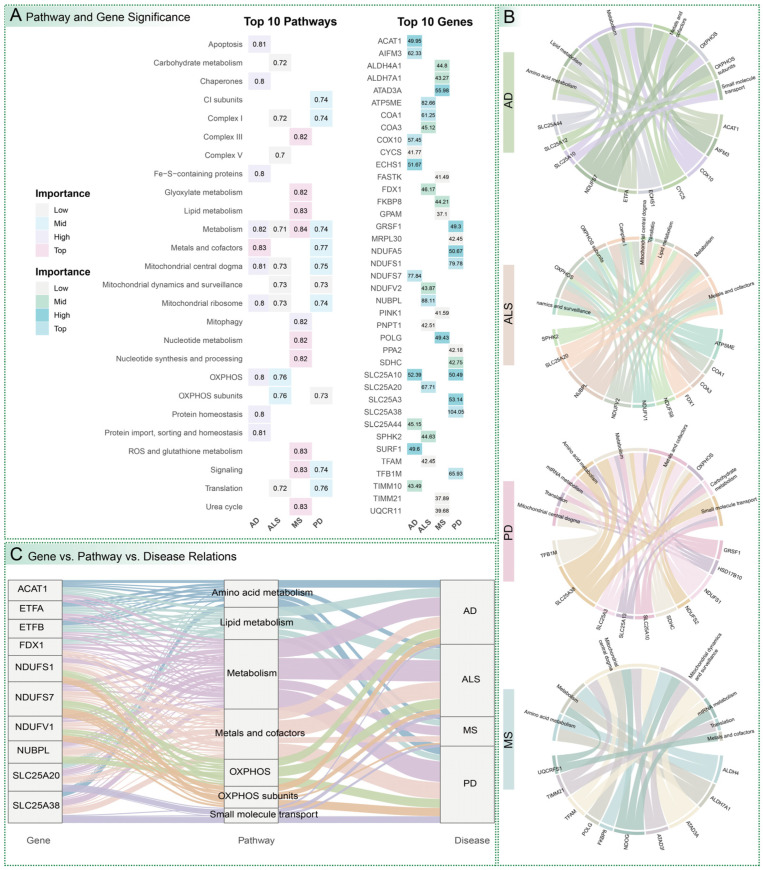
Identification and interplay of key mitochondrial pathways and genes in neurological disease prediction. This figure illustrates the most influential mitochondrial pathways and genes for predicting each neurological disease, along with their complex interrelationships. (**A**) A composite heatmap and bar chart are presented to identify the top predictive pathways and genes for AD, ALS, MS, and PD. The heatmap on the left displays the top 10 most predictive mitochondrial pathways for each disease, with color intensity and numerical values representing the composite “Pathway Score.” The bar chart on the right shows the corresponding top 10 most predictive genes, where the length and numerical value of each bar indicate the “Total Importance” score. Disease-specific patterns are revealed; for example, mitochondrial metal ion and cofactor homeostasis is highly predictive for both AD and PD, while oxidative phosphorylation (OXPHOS) is most prominent for ALS. Profound, disease-specific importance is shown by genes such as *NUBPL* (Importance: 88.11) for ALS and *SLC25A38* (Importance: 104.05) for PD. (**B**) Chord diagrams are used to illustrate the relationships between the top 10 predictive genes (outer circle, left) and their associated high-scoring pathways (outer circle, right) for each of the four diseases. The width of the connecting ribbons (chords) is proportional to the strength of each gene’s contribution within its pathway. These diagrams provide a visual representation of how key genes are functionally distributed within mitochondrial pathways. For instance, in AD, genes such as *NDUFS7* and *COX10* are strongly linked to OXPHOS pathways, while in PD, *SLC25A38* is heavily connected to pathways involving small molecule transport and metal ion homeostasis. (**C**) A Sankey diagram illustrates the cross-disease relationships between a selection of the most influential genes (left column), their primary associated mitochondrial pathways (middle column), and the diseases in which they are most influential (right column). The flow and thickness of the bands visualize how certain core pathways, such as metals and cofactors, as well as OXPHOS, are predictive across multiple diseases but are driven by different key genes in each context. This diagram highlights both shared pathogenic themes (e.g., the importance of OXPHOS) and disease-specific molecular drivers (e.g., the exceptional importance of *SLC25A38* for PD and *NUBPL* for ALS, as well as the broader metabolic gene involvement in AD and MS).

**Figure 5 cimb-47-00636-f005:**
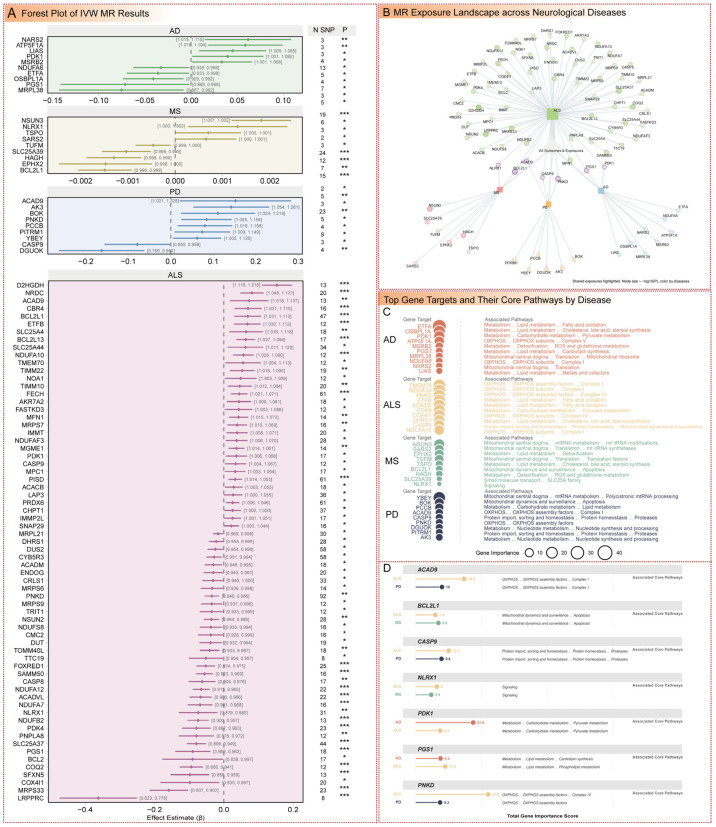
Mendelian randomization analysis of causal gene associations in neurological diseases. The causal relationships between genetically predicted mitochondrial gene expression and the risk for four neurological diseases were investigated using MR. (**A**) Forest plots of the Inverse Variance Weighted (IVW) MR results, showing the effect estimates (b) with 95% confidence intervals (CI) for selected genes associated with each disease. In AD, genes such as *LIAS* (OR 1.046, 95% CI 1.009–1.085, *p* = 0.015), *NARS2* (OR 1.064, 95% CI 1.015–1.115, *p* = 0.009), and *ATP5F1A* (OR 1.061, 95% CI 1.019–1.104, *p* = 0.004) were associated with increased risk, suggesting a role in lipoic acid synthesis, mitochondrial translation, and ATP synthase function. Conversely, *PGS1* (OR 0.931, 95% CI 0.868–0.998, *p* = 0.045), ETFA (OR 0.965, 95% CI 0.933–0.998, *p* = 0.038), and *MRPL38* (OR 0.927, 95% CI 0.867–0.992, *p* = 0.029) were protective, playing roles in cardiolipin synthesis and mitochondrial ribosomal function. In ALS, *NRDC* (OR 1.084, 95% CI 1.048–1.122, *p* = 2.89 × 10^−6^), *BCL2L13* (OR 1.065, 95% CI 1.037–1.094, *p* = 3.41 × 10^−6^), and *D2HGDH* (OR 1.165, 95% CI 1.116–1.216, *p* = 2.50 × 10^−12^) were associated with increased risk, implicating protein processing, apoptosis regulation, and organic acid metabolism in motor neuron degeneration. Protective genes in ALS included MRPS33 (OR 0.854, 95% CI 0.807–0.902, *p* = 2.41 × 10^−8^) and *LRPPRC* (OR 0.696, 95% CI 0.623–0.778, *p* = 1.76 × 10^−10^), highlighting roles in mitochondrial ribosomal and RNA processing. In MS, smaller effect sizes were observed for *NSUN3* (OR 1.002, 95% CI 1.001–1.002, *p* = 1.42 × 10^−8^) and *NLRX1* (OR 1.001, 95% CI 1.000–1.002, *p* = 0.014), while genes like *SLC25A39* (OR 0.999, 95% CI 0.998–0.999, *p* = 9.12 × 10^−5^) and *BCL2L1* (OR 0.999, 95% CI 0.998–0.999, *p* = 2.80 × 10^−7^) were protective. In PD, *PITRM1* (OR 1.077, 95% CI 1.009–1.149, *p* = 0.026) and *ACAD9* (OR 1.164, 95% CI 1.021–1.328, *p* = 0.023) increased risk, indicating an association with mitochondrial protein degradation and electron transport chain biogenesis, while *DGUOK* (OR 0.850, 95% CI 0.769–0.940, *p* = 0.001) and *CASP9* (OR 0.924, 95% CI 0.855–0.998, *p* = 0.045) were protective, implicating mitochondrial DNA synthesis and apoptosis regulation. The asterisks in panel A denote the level of statistical significance as follows: * *p* < 0.05, ** *p* < 0.01, and *** *p* < 0.001. (**B**) The MR exposure landscape visualizes the distribution and associations of genes and pathways across the diseases, highlighting shared and disease-specific relationships. For example, *PGS1* was identified as a protective factor for both AD (OR 0.931, 95% CI 0.868–0.998, *p* = 0.045) and ALS (OR 0.914, 95% CI 0.869–0.962, *p* = 5.00 × 10^−4^), involved in cardiolipin synthesis, while *PDK1* was linked to increased risk in both AD (OR 1.041, 95% CI 1.001–1.082, *p* = 0.042) and ALS (OR 1.037, 95% CI 1.006–1.068, *p* = 0.019), related to pyruvate metabolism. (**C**) Top gene targets and their associated core pathways are illustrated for each disease. Genes such as *ACAD9* (ALS: Total Importance 18.40, PD: Total Importance 9.97) and *PNKD* (ALS: Total Importance 27.55, PD: Total Importance 9.26) were linked to OXPHOS assembly factors in ALS and PD, with *PNKD* showing reduced risk in ALS (OR 0.966, 95% CI 0.946–0.986, *p* = 0.001) but increased risk in PD (OR 1.088, 95% CI 1.025–1.154, *p* = 0.005). *CASP9* (ALS: Total Importance 12.49, PD: Total Importance 9.80) was protective in ALS (OR 1.035, 95% CI 1.004–1.067, *p* = 0.028) but showed increased risk in PD (OR 0.924, 95% CI 0.855–0.998, *p* = 0.045), reflecting its role in protein homeostasis and apoptosis regulation. (**D**) Total gene importance scores are presented for the most influential genes, such as *ACAD9* (ALS: 18.40, PD: 9.97), *BCL2L1* (MS: 8.62, ALS: 7.38), and *PGS1* (AD: 9.43, ALS: 11.21), showing their contribution to disease-specific pathways. These findings highlight the shared and context-dependent roles of mitochondrial genes in the risk of neurological diseases, with certain genes influencing multiple diseases through similar or distinct pathways. Throughout the figure, different colors are used to represent the four neurological diseases.

**Figure 6 cimb-47-00636-f006:**
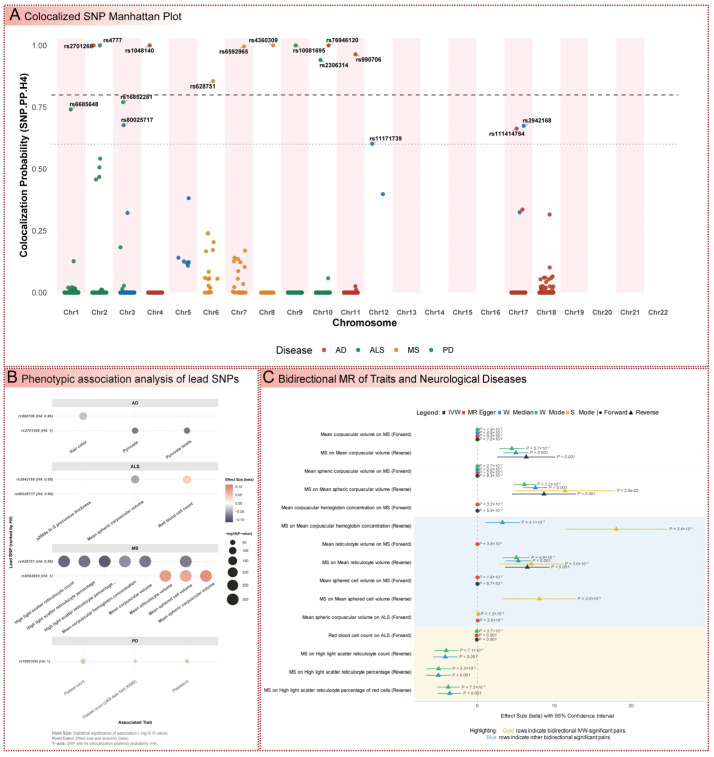
Colocalization analysis, phenotypic associations, and bidirectional Mendelian randomization of mitochondrial pathway-related genes in neurological diseases. The integration of colocalization analysis, PheWAS, and bidirectional MR to explore the genetic links between mitochondrial pathway-related genes and neurological diseases. (**A**) Colocalization of eQTLs with disease risk variants is shown in the Manhattan plot. For AD, SNPs such as rs76946120, rs2701268, and rs1048140 exhibited a colocalization probability (SNP.PP.H4) of 1.00, indicating strong evidence of shared causal variants. In MS, SNPs rs4360309 (SNP.PP.H4 1.00) and rs6592965 (SNP.PP.H4 0.996) also showed near-certain colocalization, suggesting the involvement of mitochondrial pathways in immune-mediated pathology. For PD, SNPs like rs10081695 and rs4777 (SNP.PP.H4 1.00) demonstrated robust colocalization, highlighting genetic links to dopaminergic neuron loss. In ALS, moderate colocalization signals were found for rs80025717 (SNP.PP.H4 0.678) and rs2942168 (SNP.PP.H4 0.675), suggesting potential shared genetic influences. (**B**) Phenotypic associations of colocalized SNPs were assessed through PheWAS. In AD, rs2701268 was significantly associated with pyruvate levels (beta −0.093, *p* = 1.80 × 10^−70^), linking mitochondrial pyruvate metabolism to cognitive decline. In ALS, rs2942168 was associated with red blood cell count (beta 0.052, *p* = 6.58 × 10^−122^) and mean spheric corpuscular volume (beta −0.060, *p* = 4.90 × 10^−128^), reflecting systemic impacts on erythrocyte characteristics. MS SNPs rs6592965 and rs628751 were linked to mean reticulocyte volume (beta 0.101, *p* = 0.00) and mean corpuscular volume (beta −0.095, *p* = 0.00), suggesting a role in red blood cell development and potential immune system modulation. For PD, rs10081695 was associated with platelet count (beta 0.067, *p* = 1.37 × 10^−49^), which may relate to non-motor symptoms of the disease. (**C**) Forest plot from bidirectional MR analysis illustrating causal relationships between phenotypic traits and neurological diseases. Key findings for MS include a protective effect of higher mean corpuscular volume (MCV) on disease risk (IVW, OR: 0.9993; 95% CI: 0.9986–0.9999; *p* = 0.016). The significant reverse association (IVW, OR: 577.67; 95% CI: 13.14–25,389.97; *p* = 9.86 × 10^−4^) should be interpreted with caution due to the limited number of instrumental variables. Similar protective effects against MS were observed for mean spheric corpuscular volume (IVW, OR: 0.9993; 95% CI: 0.9987–0.9998; *p* = 0.009) and, in a sensitivity analysis, for mean reticulocyte volume (MR-Egger, OR: 0.9985; 95% CI: 0.9972–0.9999; *p* = 0.038). For ALS, a sensitivity analysis linked higher mean spheric corpuscular volume to increased disease risk (MR-Egger, OR: 1.042; 95% CI: 1.005–1.080; *p* = 0.025), whereas a robust protective effect was found for red blood cell count (IVW, OR: 0.930; 95% CI: 0.901–0.959; *p* = 3.75 × 10^−6^). These results highlight the complex genetic relationships between mitochondrial pathways and neurological diseases, providing further insights into the mechanisms driving disease risk and informing therapeutic strategies.

**Figure 7 cimb-47-00636-f007:**
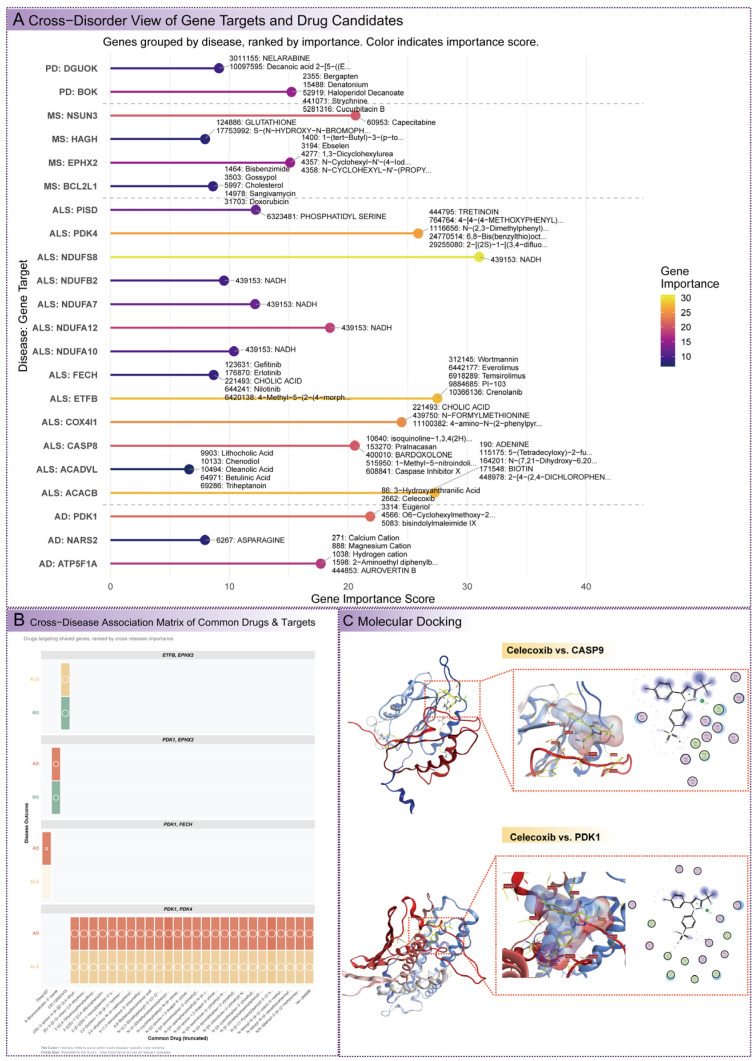
Drug enrichment analysis and molecular docking of multi-disease targets. The identification of potential drug candidates targeting mitochondrial pathways associated with neurological diseases, along with molecular docking simulations to evaluate binding affinity. (**A**) The cross-disease view of gene targets and drug candidates illustrates key genes grouped by disease, ranked by importance. Mitochondrial genes and associated drugs are color-coded by gene importance. For Alzheimer’s Disease (AD), genes such as *ATP5F1A* and *PDK1* are prioritized, with drugs like Aurovertin B and Celecoxib showing potential therapeutic interactions. In Amyotrophic Lateral Sclerosis (ALS), genes like *ACACB* and *ETFB* are targeted by drugs such as Biotin and Wortmannin, respectively. The gene importance scores reflect their relevance across diseases, with shared targets like *PDK1* emerging for both AD and ALS. (**B**) The cross-disease association matrix presents common drugs and their gene targets, highlighting the relationships between drug efficacy and gene interactions across AD, ALS, and Multiple Sclerosis (MS). For example, *PDK1* and *PDK4* are targeted by compounds such as *N*-(2,3-Dimethylphenyl)-2-((3-(2-hydroxyphenyl)-1H-1,2,4-triazol-5-yl)thio)acetamide, showing significant importance in both AD and ALS, indicating shared vulnerabilities in mitochondrial pyruvate metabolism. (**C**) Molecular docking simulations of Celecoxib with two key targets—*CASP9* (in PD) and *PDK1* (in AD and ALS)—are depicted. The binding affinities, expressed through docking scores, suggest that Celecoxib binds more strongly to *PDK1* (dock score of −6.52) compared to CASP9 (dock score of −5.47), supporting its higher therapeutic potential in modulating pyruvate metabolism in AD and ALS. These findings underscore Celecoxib’s multi-target capabilities, positioning it as a promising candidate for addressing mitochondrial dysfunction across these diseases.

**Table 1 cimb-47-00636-t001:** Summary of GWAS and gene expression studies in neurological diseases.

Disease	GWAS ID	Sample Size (Case/Control)	Description	PMID
**Alzheimer’s Disease (AD)**	ebi-a-GCST90027158	39,106 clinically diagnosed cases, 46,828 proxy cases, 401,577 controls	A two-stage GWAS identified 75 AD risk loci, 42 of which are novel. Pathway analyses revealed amyloid/tau involvement and microglia implications, with a genetic risk score showing a 1.6- to 1.9-fold increased AD risk across deciles.	35379992
**Amyotrophic Lateral Sclerosis (ALS)**	ebi-a-GCST005647	20,806 ALS cases, 59,804 controls	A GWAS identified *KIF5A* mutations in the C-terminal domain as a novel ALS risk factor, implicating cytoskeletal defects. The study contrasted these with SPG10 and CMT2 mutations, linking them to ALS pathogenesis.	29566793
**Multiple Sclerosis (MS)**	ukb-b-17670	462,933 (1679 cases, 461,254 controls)	A large-scale GWAS from the UK Biobank analyzed genetic factors contributing to MS risk.	--
**Parkinson’s Disease (PD)**	ieu-b-7	482,730 (33,674 cases, 449,056 controls)	The International Parkinson’s Disease Genomics Consortium conducted a GWAS to investigate genetic factors in PD.	--
**Disease**	GEO NO.	Sample Size (case/control)	Description	PMID
**AD**	GSE118553	33 AsymAD, 52 AD, 27 Control	Differential and co-expression analysis on brain tissue samples identified significant transcriptomic changes in the frontal cortex of AsymAD subjects. A total of 14 genes, including *GPM6B* and *ANKEF1*, were linked to AD neuropathology.	31063847
**ALS**	GSE112681	396 ALS patients, 75 ALS mimic diseases, 645 healthy controls	Microarray analysis identified 752 ALS-increased and 764 ALS-decreased DEGs, highlighting gene expression shifts resembling acute stress responses. A 61-gene signature was linked to improved survival prediction.	29939990/ 31118040
**MS**	GSE131282	64 MS NAGM samples, 42 control gray matter samples	Gene expression microarray analysis found elevated *HLA-DRB1* expression in MS NAGM, especially in cases with the HLA-DR15 haplotype, suggesting a role in MS pathogenesis.	31882398
**PD**	GSE28894	14–15 control brains, 11–15 PD brains across four regions	Gene expression profiling identified modest differences between PD and control brains. RNA was generated from 500 ng of total RNA from these regions (medulla: 15 control brains, 14 PD brains; striatum: 15 control brains, 15 PD brains; frontal cortex: 15 control brains, 11 PD brains; cerebellum: 14 control brains, 15 PD brains).	--

## Data Availability

The datasets used in this study are publicly available and can be accessed through the respective data repositories. Genome-wide association study (GWAS) summary statistics for Alzheimer’s Disease (AD), Amyotrophic Lateral Sclerosis (ALS), Multiple Sclerosis (MS), and Parkinson’s Disease (PD) can be accessed through the OpenGWAS database (https://gwas.mrcieu.ac.uk/, accessed on 5 August 2025). Gene expression data from the microarray datasets (GSE118553 for AD, GSE112681 for ALS, GSE131282 for MS, and GSE28894 for PD) are available on the Gene Expression Omnibus (GEO) platform (https://www.ncbi.nlm.nih.gov/geo/, accessed on 5 August 2025). Additionally, mitochondrial pathway data from the MitoCarta 3.0 database (https://www.broadinstitute.org/mitocarta/mitocarta30-inventory-mammalian-mitochondrial-proteins-and-pathways, accessed on 5 August 2025) can be obtained from the provided resources. All data are available for download and were used in accordance with their respective data use policies. Further details on the specific datasets and access instructions are provided in the Appendix A.

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
