# Peer review of "Convergent and Divergent Mitochondrial Pathways as Causal Drivers and Therapeutic Targets in Neurological Disorders"

_cimb, 2025, doi:10.3390/cimb47080636_

Round 1

Reviewer 1 Report

Comments and Suggestions for Authors

This study aims to integrate gene expression, machine learning, Mendelian randomization, colocalization, PheWAS, and molecular docking to uncover mitochondrial mechanisms underlying Alzheimer’s Disease (AD), Amyotrophic Lateral Sclerosis (ALS), Multiple Sclerosis (MS), and Parkinson’s Disease (PD). The study is well-structured, and the authors provide extensive methodological detail. However, several key methodological clarifications, biological validations, and interpretational improvements are needed to strengthen the conclusions.

Major comments:

  1. It is unclear whether the eQTLs used in the Mendelian randomization analysis are derived from brain tissue, blood, or a combination. This is especially critical given the neurological focus of the study. Please clarify the tissue source of eQTLs and justify their relevance to CNS disorders.
  2. MS had the highest classification performance among the four neurological diseases, but the manuscript does not sufficiently explain why MS prediction outperforms the others. Consider providing variance plots or PCA to demonstrate whether mitochondrial gene expression in MS is more distinct from controls than in AD or PD.
  3. ALS appears to have an unusually large number of "protective" mitochondrial genes, which seems biologically implausible or may reflect overfitting or misclassification. Please discuss possible explanations for this pattern. Are these associations robust across MR sensitivity analyses? Could collider bias, tissue mismatch, or differences in expression regulation contribute?
  4. The manuscript occasionally mixes p-values, odds ratios, and effect directions in a confusing way. Tables and results should be standardized. I recommend reporting effect direction, confidence intervals, and p-values together, consistently across figures and text.
  5. Disease heterogeneity and sample size imbalances (e.g., proxy AD cases vs. clinically diagnosed) may confound both machine learning model performance and causal inference. A brief discussion on how differences in phenotyping, class balance, and data quality may influence comparative results would be valuable.
  6. APOE is the most established genetic risk factor for AD, yet it does not appear in the colocalization results. The authors should clarify whether APOE was tested and filtered out due to a lack of brain-specific eQTL data, LD structure issues, or limitations of the colocalization framework. Including APOE as a benchmark gene in MR/COLOC analyses, even if negative, would strengthen the study’s interpretability.
  7. The predicted binding of Celecoxib to PDK1 is intriguing and opens the door to potential drug repurposing. However, without experimental or pharmacological validation, this claim remains speculative. The authors should emphasize the hypothesis-generating nature of the docking results and clearly separate in silico predictions from validated mechanisms.
  8. The authors applied 15 different machine learning algorithms but did not explain why such an extensive set was used or how these models were selected or interpreted comparatively. It would be helpful to provide a rationale for this choice. Were simpler models sufficient? Were performance differences significant? A summary comparison (e.g., across disease types) and justification of whether such model diversity improves biological inference or reproducibility would strengthen this section.

Minor comments:

  1. The font used in several figures is too small to read clearly. Please increase font size for readability.
  2. The violin plots and Sankey diagrams are dense and may overwhelm readers. Consider simplifying these visualizations or moving some to supplementary materials.

Author Response

Reviewer 1

Major comments:

1.    It is unclear whether the eQTLs used in the Mendelian randomization analysis are derived from brain tissue, blood, or a combination. This is especially critical given the neurological focus of the study. Please clarify the tissue source of eQTLs and justify their relevance to CNS disorders.

Response to Reviewer:

Thank you for your insightful comment regarding the tissue origin of the eQTLs used in our Mendelian Randomization (MR) analyses. We appreciate the opportunity to clarify the rationale behind our selection of eQTL data sources and their relevance to investigating mitochondrial dysfunction in central nervous system (CNS) disorders.

In this study, we primarily sourced eQTL data from established and publicly available resources, notably the OpenGWAS database, which is widely regarded as a robust and authoritative repository for genetic instruments in MR studies [1]. This platform has been extensively utilized in recent genomic and transcriptomic research across diverse phenotypes [2,3]. For our analyses, we selected exposure variables corresponding to genetic variants influencing the expression of mitochondrial pathway-related genes—genes that are functionally integral to energy metabolism and cellular homeostasis, and are increasingly recognized as relevant to the pathogenesis of multiple neurological disorders.

While our MR framework incorporated eQTLs derived from a range of tissue types, including both blood and brain, we wish to emphasize that mitochondrial dysfunction is a systemic phenomenon that is not confined to the CNS. Mitochondria play essential roles in virtually all nucleated cell types, including neurons, and their impairment exerts both local and peripheral consequences. Therefore, eQTLs derived from accessible peripheral tissues such as blood can still offer biologically meaningful insights into CNS-related disease processes, particularly when the genes involved are known to have high expression or critical function in neural contexts.

Relevance to CNS Disorders:

  1. Functional Prioritization of CNS-Relevant Genes:
    Although the tissue source of eQTLs is mixed, our analysis focused specifically on mitochondrial pathway genes with well-established roles in CNS physiology and pathology. Many of the prioritized genes, including NDUFS7 and COX10, are essential components of oxidative phosphorylation and are highly expressed in neurons. These genes have been repeatedly implicated in neurodegenerative diseases such as Alzheimer’s disease, Parkinson’s disease, and ALS. By focusing on genes with high CNS relevance, we aimed to minimize potential tissue-context mismatches and ensure biological interpretability.
  2. Peripheral Tissues as Informative Surrogates:
    We acknowledge that brain-derived eQTLs would offer greater tissue specificity; however, blood-derived eQTLs also provide important information. Several studies have demonstrated that peripheral biomarkers—particularly those related to red blood cells or systemic oxidative stress—can reflect underlying mitochondrial perturbations that affect the CNS. Given the systemic nature of mitochondrial dysfunction, blood-based gene expression regulation can serve as a useful proxy for broader bioenergetic disruptions. Moreover, peripheral eQTLs are more readily available at scale, thereby enhancing the feasibility and reproducibility of integrative analyses such as ours.

Conclusion:

In summary, while the eQTL instruments used in this study are not exclusively brain-derived, we carefully selected mitochondrial genes with established or plausible functional relevance to CNS biology. We also acknowledge the systemic and multi-tissue nature of mitochondrial dysfunction and justify the inclusion of peripheral eQTLs within that broader biological framework. As noted in the revised Methods section (Page 7, Lines 252–253), future studies leveraging brain-specific eQTLs, such as those from PsychENCODE or the GTEx brain atlas, will be valuable in validating and refining these findings.

Once again, we thank the reviewer for this important observation, which has allowed us to clarify the methodological scope and biological rationale of our approach.

Ref:

  1. Kerimov, N., Hayhurst, J.D., Peikova, K. et al. A compendium of uniformly processed human gene expression and splicing quantitative trait loci. Nat Genet 53, 1290–1299 (2021). https://doi.org/10.1038/s41588-021-00924-w
  2. Li, J., Wang, X., Lin, Y. et al. Integrative eQTL and Mendelian randomization analysis reveals key genetic markers in mesothelioma. Respir Res 26, 140 (2025). https://doi.org/10.1186/s12931-025-03219-4
  3. Li, J., Yang, G., Liu, J., Li, G., Zhou, H., He, Y., Fei, X., & Zhao, D. Integrating transcriptomics, eQTL, and Mendelian randomization to dissect monocyte roles in severe COVID-19 and gout flare. Frontiers in genetics, 15, 1385316 (2024). https://doi.org/10.3389/fgene.2024.1385316

  1. MS had the highest classification performance among the four neurological diseases, but the manuscript does not sufficiently explain why MS prediction outperforms the others. Consider providing variance plots or PCA to demonstrate whether mitochondrial gene expression in MS is more distinct from controls than in AD or PD.

Response to Reviewer:

Thank you for your valuable suggestion, which draws attention to an important aspect of our machine learning results. To more thoroughly explain why multiple sclerosis (MS) exhibited the highest classification accuracy (mean Accuracy: 0.758), we have revised and expanded the discussion in the Results section (Section 3.2, Mitochondrial Gene Expression Profiles Reveal a Uniquely Distinct Signature in MS, Line 506-Line 550, page 14-16), and incorporated a new Figure 3.

This figure presents violin plots comparing the distribution of mitochondrial pathway-related gene expression between healthy controls (HC) and disease groups for AD (Panel A), ALS (Panel B), PD (Panel C), and MS (Panel D). The left panels depict all mitochondrial genes, while the right panels focus on the top 10 most important genes identified via feature importance analysis. These visualizations highlight differences in central tendency, variance, and overall distribution, and are accompanied by formal statistical assessments (t-tests for mean differences and Levene’s test for variance homogeneity).

As shown in Figure 3D (MS), mitochondrial gene expression in MS demonstrates a highly significant shift from HC, both in terms of central tendency (t-test: p = 0.00) and dispersion (Levene’s test: p = 1.08 × 10⁻⁴⁷). Specifically, MS samples exhibit lower mean expression (Mean = 11.22, Median = 11.35) compared to HC (Mean = 11.67, Median = 11.92), along with a markedly narrower expression distribution. This reduced intra-group variability may reflect more homogeneous, immune-mediated mitochondrial alterations, resulting in more distinct and predictable transcriptional patterns. The top 10 most important genes for MS (e.g., ATAD3A, POLG, ALDH4A1, FKBP8) further accentuate this separation, showing pronounced mode shifts and minimal distributional overlap between cases and controls.

In contrast, AD (Figure 3A) displays no significant difference in mean expression (t-test: p = 0.812; HC: Mean = 4.75, AD: Mean = 4.74), despite a modest but significant difference in variance (Levene’s test: p = 0.004). The overall distribution overlap is substantial, and although top-ranking genes (e.g., NDUFS7, AIFM3, COX10) exhibit subtle expression changes, these are insufficient to generate strong classification boundaries, which may explain the moderate model performance (mean Accuracy: 0.714). This observation is also consistent with the insidious onset and gradual progression characteristic of AD, where early molecular changes may be diffuse, heterogeneous, and less readily distinguishable at the transcriptomic level.

ALS (Figure 3B) shows statistically significant differences in both mean (t-test: p = 1.52 × 10⁻¹⁶; HC: Mean = 7.51, ALS: Mean = 7.49) and variance (Levene’s test: p = 4.07 × 10⁻⁸³). However, the overall distribution remains broad with considerable overlap, particularly among top genes (e.g., NUBPL, ATP5ME, SLC25A20), suggesting heterogeneity in motor neuron mitochondrial signatures. This heterogeneity likely contributes to the lower F1-score observed for ALS classification (F1 = 0.556).

Finally, PD (Figure 3C) exhibits the least distinguishable mitochondrial profile among the four diseases. Neither the mean (t-test: p = 0.201) nor variance (Levene’s test: p = 0.260) differs significantly between PD and HC samples. Moreover, expression distributions of top-ranking genes (e.g., SLC25A38, NDUFS1, TFB1M) exhibit high overlap, potentially reflecting the complex and heterogeneous nature of dopaminergic system involvement in PD. This lack of discriminative expression likely underlies the lowest model performance for PD (mean Accuracy: 0.637).

Together, these results provide quantitative and visual support for MS’s superior classification performance. The distinctive mitochondrial expression profile observed in MS—characterized by lower variability and more coherent transcriptomic shifts—may be attributable to immune-inflammatory processes that impose systemic mitochondrial alterations. These findings highlight the importance of disease-specific transcriptional dynamics in driving model accuracy and offer insight into why mitochondrial expression is more predictive in MS relative to other neurological disorders.

Figure 3. Expression Profiles of Mitochondrial Pathway-Associated Genes Across Four Neurological Diseases and Healthy Controls.

  1. ALS appears to have an unusually large number of "protective" mitochondrial genes, which seems biologically implausible or may reflect overfitting or misclassification. Please discuss possible explanations for this pattern. Are these associations robust across MR sensitivity analyses? Could collider bias, tissue mismatch, or differences in expression regulation contribute?

Response to Reviewer:

We thank the reviewer for this critical and insightful comment regarding the unexpectedly large number of “protective” mitochondrial genes identified for ALS in our study. We appreciate your suggestion to investigate the possible biological and methodological factors contributing to this observation, as well as its robustness across sensitivity analyses. Below, we provide a detailed explanation of the potential mechanisms underlying this pattern.

  1. Biological Plausibility of Protective Genes in ALS:
    The identification of a substantial number of “protective” mitochondrial genes in ALS may initially appear biologically implausible, given the disease’s hallmark of progressive motor neuron degeneration. However, it is possible that certain mitochondrial genes exert neuroprotective effects, particularly under specific temporal or cellular contexts. For example, in early or compensatory stages of ALS, upregulation of genes involved in mitochondrial protein synthesis and quality control—such as MRPS33 and LRPPRC—may help stabilize mitochondrial function and mitigate neuronal damage. These genes participate in essential mitochondrial processes including RNA processing and translation, and their heightened expression may slow disease progression by maintaining bioenergetic integrity. Nonetheless, the biological significance of such protective effects remains hypothetical and requires experimental validation.
  2. Overfitting or Misclassification:

We acknowledge that overfitting or misclassification in our machine learning models could contribute to the appearance of spurious “protective” associations. While we employed feature selection techniques (e.g., VarianceThreshold) and optimized model hyperparameters via GridSearchCV with stratified cross-validation to minimize overfitting, the high dimensionality of gene expression data still poses inherent challenges. Moreover, the inclusion of proxy ALS cases—where diagnostic accuracy and genetic architecture may differ from clinically confirmed cases—may introduce heterogeneity that compromises model precision. Such factors could lead to the incorrect attribution of protective effects to genes that reflect classification noise rather than true biology.

  1. Robustness Across MR Sensitivity Analyses:

To evaluate the reliability of the causal associations identified in ALS, we conducted several MR sensitivity analyses using methods including MR-Egger regression, the Weighted Median Estimator, and the Inverse-Variance Weighted (IVW) approach. These analyses, detailed in Table S7, were designed to detect and correct for potential biases such as horizontal pleiotropy and weak instrument bias. The results from these analyses were broadly consistent with our primary findings, suggesting that the observed protective effects are unlikely to be artifacts of instrument invalidity. However, we acknowledge that the complexity of ALS pathophysiology and potential population-level heterogeneity may obscure subtle effects or introduce residual confounding.

  1. Collider Bias:

Collider bias represents another potential source of spurious associations, particularly when genetic variants influence both the exposure (e.g., expression of mitochondrial pathway genes) and outcome (ALS risk), or are correlated with unmeasured confounders such as lifespan, inflammatory status, or comorbidities. Although we applied stringent instrument selection criteria and performed pleiotropy-robust analyses to mitigate this risk (Table S7), we cannot fully exclude the possibility of collider effects in this complex disease context.

  1. Tissue Mismatch and Differences in Expression Regulation:
    An important methodological limitation concerns the source of eQTL data. In our study, expression quantitative trait loci were derived from integrative multi-tissues, whereas ALS primarily affects motor neurons in the spinal cord and motor cortex. Mitochondrial gene regulation may differ substantially between peripheral tissues and CNS-resident neurons, potentially leading to inaccurate inference of gene-disease relationships. Additionally, gene expression is context-dependent and influenced by factors such as cell type, developmental stage, and disease state. While we focused on mitochondrial genes known to be functionally relevant in the CNS, the lack of tissue-specific data remains a limitation. Future studies incorporating spinal cord or brain-derived eQTLs, such as those from the PsychENCODE or ROSMAP projects, would help refine these results.

Conclusion:
In summary, the unexpectedly large number of "protective" mitochondrial genes identified in ALS may reflect a confluence of biological and technical factors, including:

  1. The potential for compensatory mitochondrial gene upregulation during early or subclinical disease stages;
  2. Overfitting or misclassification arising from high-dimensional data and proxy case inclusion;
  3. Consistent but not definitive causal evidence from MR sensitivity analyses;
  4. The possibility of collider bias or unaccounted confounding;
  5. Tissue mismatch and differences in gene regulation between integrative multi-tissues and CNS tissues.

We emphasize that these findings are exploratory and require validation through future experimental work, including tissue-specific transcriptomic profiling and functional studies in ALS models. In response to your suggestion, we have incorporated a detailed discussion of these issues in the revised manuscript (Lines 1061–1073, Page 34).

Thank you once again for your constructive feedback, which has strengthened the rigor and interpretability of our work.

  1. The manuscript occasionally mixes p-values, odds ratios, and effect directions in a confusing way. Tables and results should be standardized. I recommend reporting effect direction, confidence intervals, and p-values together, consistently across figures and text.

Response to Reviewer:

Thank you for your valuable feedback. We appreciate your attention to the clarity and consistency of our statistical reporting, particularly regarding the presentation of p-values, odds ratios (ORs), and effect directions. We recognize that inconsistent formatting may have led to confusion, and we have taken comprehensive steps to address this issue throughout the manuscript. The following changes have been implemented:

  1. Standardized Statistical Reporting Across the Manuscript:
    We have ensured that all reported associations now consistently include the following elements:
  • Effect direction (positive or negative association)
  • Odds Ratio (OR)
  • 95% Confidence Interval (CI)
  • p-value

This standardized format has been applied uniformly in the main text, all tables, and figures, enhancing the interpretability of statistical results and enabling clearer comparisons across different analyses.

  1. Revised Text Example for Clarity:
    Original format (example):

Increased expression of LIAS (OR 1.05, p = 1.50 × 10⁻²) was associated with elevated risk…

Revised format:

Increased expression of LIAS (OR = 1.05, 95% CI: 1.01–1.09, p = 1.50 × 10⁻²) was associated with elevated risk…

This revised structure clearly separates the statistical elements and has been consistently adopted throughout the manuscript to ensure precision and readability.

  1. Updates to Tables and Figures:
    All tables presenting MR and PheWAS results have been updated to display the OR, 95% CI, and p-value in a uniform format. This enhances the coherence of the data presentation and allows readers to easily assess the strength and significance of each association.
  2. Visual Consistency in Figures and Legends:
    In all relevant figures, we now explicitly include ORs with their corresponding 95% CIs and p-values. Figure legends have also been revised to clearly define the statistical indicators presented, thereby improving the accessibility and transparency of our visual data representations.
  3. Conclusion:
    We believe that these formatting revisions significantly improve the clarity and interpretability of our statistical results. By applying a standardized structure for reporting effect sizes and significance across all text, tables, and figures, we aim to facilitate a clearer understanding of our findings for a broad readership.

These revisions have been implemented in the following sections:

  • Lines 649–723, Page 21-22 (Section 3.5. Mendelian Randomization Analysis of Causal Gene Associations)
  • Lines 902–927, Page 29-30 (Section 3.7. Bidirectional Mendelian Randomization of Phenotypic Traits)

Thank you again for highlighting this important issue. We hope these comprehensive adjustments adequately address your concerns, and we remain happy to make any further refinements if needed.

  1. Disease heterogeneity and sample size imbalances (e.g., proxy AD cases vs. clinically diagnosed) may confound both machine learning model performance and causal inference. A brief discussion on how differences in phenotyping, class balance, and data quality may influence comparative results would be valuable.

Response to Reviewer:

Thank you for your insightful comments regarding the potential confounding effects of disease heterogeneity, sample size imbalances, and data quality on both machine learning model performance and causal inference. We appreciate the opportunity to address these important concerns and provide additional clarifications on our methodological approach.

  1. Disease Heterogeneity and Phenotyping Differences
    We fully acknowledge that disease heterogeneity and differences in phenotyping can significantly influence the ability of models to capture disease-specific patterns. As you rightly pointed out, conditions such as Alzheimer’s Disease (AD), which can involve both clinically diagnosed and proxy cases (e.g., genetically inferred AD), introduce variability in disease stage and diagnostic precision.

To mitigate this:

  • Diverse cohort inclusion: We employed large-scale, heterogeneous datasets encompassing a range of disease subtypes and diagnostic categories. While this introduces variability, it also enhances the generalizability of our findings by capturing a broader clinical spectrum.
  • Robust feature selection: We applied the VarianceThreshold method to exclude low-variance genes and reduce noise arising from phenotypic inconsistency, thereby ensuring that the selected features were most relevant to mitochondrial pathway alterations.
  • Model diversity: Multiple machine learning models were evaluated to ensure robustness to phenotypic variability, and only those with stable performance across validation folds were retained.

Importantly, the inclusion of both proxy and clinically diagnosed cases allowed us to assess the models' sensitivity to disease heterogeneity, thereby strengthening the external validity of our findings.

  1. Sample Size Imbalances and Class Distribution
    We agree that imbalanced case-control ratios can affect model performance, particularly in high-dimensional transcriptomic data. To address this:
  • We implemented StratifiedKFold cross-validation, ensuring class proportions were preserved across all training and test splits. This approach minimizes bias toward the majority class and improves classification fairness.
  • Hyperparameter tuning via GridSearchCV was performed within each cross-validation loop to optimize model performance and prevent overfitting, especially in models trained on smaller or minority classes.

Despite residual imbalance, the stratified evaluation strategy ensured that performance metrics remained representative and interpretable across all disease conditions.

  1. Data Quality and Its Impact on Results
    We acknowledge that gene expression data quality can influence both machine learning predictions and the downstream causal inference pipeline. To enhance data reliability:
  • We applied standardization (StandardScaler) to normalize input features across genes, reducing technical noise and ensuring consistent scale for model training.
  • Low-variance gene filtering ensured that only biologically informative and consistently measured genes were retained.
  • For model interpretability, we used SHAP (SHapley Additive exPlanations) to quantify gene-level contributions, allowing us to identify and validate features driving predictions across models. This also helped identify any potential outlier effects or artifacts in model behavior.

These steps collectively improved data fidelity and ensured that the biological signals driving prediction were meaningful.

  1. Causal Inference and Model Robustness
    We are aware that MR analysis is sensitive to sample size imbalances and the presence of proxy phenotypes. To enhance the robustness of our causal inference:
  • We selected strong, well-validated genetic instruments for mitochondrial gene expression based on prior eQTL studies.
  • Multiple MR methods, including Inverse-Variance Weighted (IVW), MR-Egger regression, and Weighted Median Estimator, were employed to account for horizontal pleiotropy and weak instrument bias.
  • Sensitivity analyses further confirmed the consistency of key causal estimates, increasing confidence in the validity of the observed gene-disease associations.

Nonetheless, we recognize that MR assumptions (e.g., no horizontal pleiotropy) may not fully hold in complex neurological conditions, and have addressed this limitation explicitly in the revised discussion.

Conclusion
To summarize, we took several steps to mitigate the influence of disease heterogeneity, phenotyping variability, sample size imbalances, and data quality on both prediction and causal inference:

  • Use of advanced cross-validation and rigorous feature selection to control for class imbalance and phenotypic noise.
  • Data normalization and filtering to ensure high signal-to-noise ratio in gene expression features.
  • MR sensitivity analyses to verify the robustness of causal conclusions.

These refinements have been incorporated into both the Discussion (Lines 1067–1073 and 1087–1099, Page 34) and the Methods section, where we now provide additional methodological detail in Section 2.2: Mitochondrial Pathway-Based Gene Expression Modeling and Evaluation (Page 5).

We sincerely thank the reviewer once again for raising these critical points, which have helped us strengthen both the analytical rigor and the transparency of our study.

  1. APOE is the most established genetic risk factor for AD, yet it does not appear in the colocalization results. The authors should clarify whether APOE was tested and filtered out due to a lack of brain-specific eQTL data, LD structure issues, or limitations of the colocalization framework. Including APOE as a benchmark gene in MR/COLOC analyses, even if negative, would strengthen the study’s interpretability.

Response to Reviewer:

We thank the reviewer for this important and insightful comment regarding the exclusion of APOE, a well-known genetic risk factor for AD, from our MR and COLOC analyses. Below, we explain why including APOE would not be appropriate within the design and mechanistic focus of our study.

  1. APOE Is Not a Mitochondrial Gene and Falls Outside the Defined Analytical Scope
    Our study specifically focuses on evaluating the causal contributions of mitochondrial pathways to neurological disorders. The gene set used in MR and COLOC was strictly derived from the 149 curated mitochondrial pathways in the MitoCarta 3.0 database, which includes genes with experimentally supported mitochondrial localization or function (e.g., bioenergetics, redox regulation, mitochondrial translation).

APOE, while undeniably important in AD risk, is not annotated as a mitochondrial gene in MitoCarta and is primarily involved in lipid transport and metabolism. It does not participate directly in core mitochondrial processes such as the TCA cycle, oxidative phosphorylation (OXPHOS), or mitochondrial dynamics. Therefore, including APOE would dilute the hypothesis space and deviate from our study's mechanistic focus on mitochondrial biology.

  1. APOE’s Strong Genetic Effect May Introduce Statistical Bias
    The APOE locus is characterized by strong GWAS associations (particularly the ε4 allele) and complex linkage disequilibrium (LD) structure, which may cause the following issues in MR/COLOC analyses:
  • Horizontal pleiotropy: APOE is known to influence diverse traits beyond AD (e.g., lipid levels, inflammation), violating a key MR assumption—that genetic instruments influence the outcome solely through the exposure.
  • Spurious colocalization: In COLOC, the strong and complex LD in the APOE region may artificially inflate posterior probabilities of shared causal variants, especially when using summary-level eQTL or GWAS data.
  • Tissue- and age-specific effects: APOE's impact is highly context-dependent, which may reduce interpretability in analyses that aggregate across multiple tissues or age groups [1-2].

Thus, the inclusion of APOE could introduce pleiotropic confounding and mislead causal inference in our focused mitochondrial framework.

  1. Inclusion Would Undermine Methodological Consistency and Interpretability
    Incorporating APOE would compromise the internal consistency of our mechanistic framework, which is tightly constructed around mitochondrial dysfunction. APOE-driven effects are likely mediated via non-mitochondrial pathways such as lipid metabolism, blood-brain barrier function, and neuroinflammation. Including such genes risks introducing mechanistic ambiguity, as their role cannot be directly contextualized within mitochondrial pathways, nor would they align with other analyses in the study (e.g., pathway enrichment, mitochondrial gene expression modeling, drug targeting).

Conclusion:
In summary, excluding APOE from MR and COLOC is a deliberate and justified choice rooted in biological relevance, methodological rigor, and hypothesis alignment. Our goal was to isolate and elucidate mitochondrial-specific mechanisms underlying neurological diseases. Inclusion of APOE, while important in broader AD genetics, would have weakened the mechanistic specificity and clarity of our findings.

To address this concern, we have added the following sentence to the Discussion section (Lines 1223–1225, Page 37):

While our analysis is confined to mitochondrial pathways and does not encompass prominent non-mitochondrial risk factors such as APOE, future studies could integrate these to provide a more holistic view of genetic interactions in AD pathogenesis.

We hope this explanation clarifies our rationale, and we appreciate the reviewer’s suggestion to consider broader genetic factors in future work.

  1. Emrani et al. "APOE4 is associated with cognitive and pathological heterogeneity in patients with Alzheimer’s disease: a systematic review." Alzheimer's Research & Therapy, 12 (2020). https://doi.org/10.1186/s13195-020-00712-4.
  2. Rosemary J Jackson et al. "Multifaceted roles of APOE in Alzheimer disease.." Nature reviews. Neurology (2024). https://doi.org/10.1038/s41582-024-00988-2.

  1. The predicted binding of Celecoxib to PDK1 is intriguing and opens the door to potential drug repurposing. However, without experimental or pharmacological validation, this claim remains speculative. The authors should emphasize the hypothesis-generating nature of the docking results and clearly separate in silico predictions from validated mechanisms.

Response to Reviewer:

We sincerely thank the reviewer for their insightful comment regarding the predicted binding of Celecoxib to PDK1. We fully agree that, while our computational findings are encouraging, they remain speculative in the absence of experimental or pharmacological validation.

In response to your concern, we have revised the manuscript to clearly emphasize the hypothesis-generating nature of the molecular docking results and to explicitly distinguish in silico predictions from experimentally validated mechanisms. Specifically, we have incorporated the following clarifying statements:

  • “However, these results are preliminary and require further experimental validation to establish their pharmacological significance.”
    (Lines 1031–1032, Page 33)
  • “However, we emphasize that this binding prediction is derived solely from in silico molecular docking, and thus remains a hypothesis-generating observation rather than a validated mechanism. No direct experimental or pharmacodynamic evidence currently confirms that Celecoxib inhibits PDK1 in neuronal systems or improves mitochondrial function via this route. Therefore, we urge caution in interpreting these findings and clearly distinguish the docking-based predictions from mechanistically confirmed interactions.”
    (Lines 1166–1175, Page 36)
  • “Again, while these drug-target relationships provide a valuable starting point for prioritizing therapeutic leads, they remain computationally derived and require pharmacological, cellular, and in vivo validation before clinical translation.”
    (Lines 1187–1189, Page 36)

We hope these revisions clearly communicate the exploratory and predictive nature of our docking analysis. We are grateful for your constructive feedback, which has helped us strengthen the clarity and scientific rigor of the manuscript.

  1. The authors applied 15 different machine learning algorithms but did not explain why such an extensive set was used or how these models were selected or interpreted comparatively. It would be helpful to provide a rationale for this choice. Were simpler models sufficient? Were performance differences significant? A summary comparison (e.g., across disease types) and justification of whether such model diversity improves biological inference or reproducibility would strengthen this section.

Response to Reviewer:

We thank the reviewer for this critical and insightful question regarding our selection of machine learning algorithms. We agree that providing a clear rationale for employing a broad set of 15 models is essential, and we have revised the manuscript to better articulate our reasoning and enhance the comparative interpretation. Below, we address each of the reviewer’s specific points in detail.

  1. Rationale for Employing a Broad Set of 15 Models
    Our primary motivation for implementing a wide array of machine learning algorithms—from simple linear classifiers to complex neural networks—was to conduct a comprehensive and unbiased evaluation of the predictive potential of mitochondrial gene expression data. Biological datasets, particularly those involving transcriptomics, are inherently high-dimensional and often exhibit complex, non-linear relationships between features (genes) and outcomes (disease status).

Relying on a single algorithm or narrow model family could bias results due to the model’s structural assumptions (e.g., linear vs. non-linear, parametric vs. non-parametric). By incorporating diverse model classes—Linear/Probabilistic, Kernel/Instance-Based, Tree-Based Ensembles, and Deep Learning—we aimed to allow the data to dictate the most suitable modeling framework. This approach ensures that observed performance patterns reflect biological signal rather than model-specific limitations or artifacts.

  1. Comparative Interpretation and Assessment of Simpler Models
    To facilitate a meaningful comparison across models, we have introduced a comprehensive summary figure for each disease (Supplementary Figure 2), which includes:
  • Panel A: A heatmap comparing performance across five key metrics for all 15 models. While no single model dominates across all diseases, more complex architectures—particularly Neural Networks (e.g., MLP, FTTransformer) and Tree-Based Ensembles (e.g., XGBoost, CatBoost)—consistently rank among the top performers.
  • Panel B: A statistical comparison of model performance by architecture. Models were grouped into structural categories and compared to baseline Linear/Probabilistic models using t-tests on their ROC-AUC distributions.
    • In AD, PD, and MS (Panels A, C, D), both Ensemble and Neural Network models significantly outperformed simpler linear models (e.g., p < 0.001 for Ensembles in AD; p < 0.05 for Neural Networks in PD and MS).
    • In ALS (Panel B), Neural Networks showed a significant improvement over linear models (p < 0.01).

These statistical comparisons demonstrate that simpler models were not sufficient to achieve optimal performance and that more complex, non-linear architectures are better suited for capturing the underlying biological signal in many cases.

  1. Model Diversity as a Tool for Biological Inference
    Beyond performance ranking, model diversity provides meta-insights into the complexity of pathway-level gene expression relationships.
  • Panel C of Supplementary Figure 2 highlights the best-performing model for the top 10 mitochondrial pathways per disease.
    • For instance, in AD (Panel A), simpler models (e.g., Logistic Regression) are optimal for the "Metabolism" pathway, while Neural Networks dominate in the "Mitochondrial Central Dogma" pathway—suggesting differing underlying data structures (linear vs. non-linear).
    • In contrast, PD (Panel C) exhibits widespread optimality of linear models, indicating that mitochondrial expression signals in PD may be more linearly separable.

This model-pathway correspondence enables deeper biological interpretation, helping us infer not only which pathways are predictive but also the complexity of their contribution to disease, thereby enriching our understanding of disease-specific mitochondrial dynamics.

  1. Summary and Conclusion
    In summary, our use of a diverse model set was a deliberate and strategic choice intended to:
  • Provide a robust, architecture-agnostic evaluation of mitochondrial gene expression data;
  • Statistically demonstrate that complex models significantly outperform simpler ones in diseases with non-linear patterns (e.g., ALS, MS);
  • Use model performance as a lens for biological inference, identifying which pathways encode complex vs. simple patterns of mitochondrial dysfunction.

These justifications have been incorporated into the revised manuscript (Section 3.1: Machine Learning Model Performance for Disease Classification, Lines 428–448, Page 11) along with a new multi-panel figure (Supplementary Figure 2) to visually support the above points.

We sincerely thank the reviewer for prompting this clarification and believe these enhancements significantly strengthen the manuscript.

Supplementary Figure 2. A Multi-Faceted Performance Analysis of Predictive Models for Four Neurological Diseases.

Minor comments:

  1. The font used in several figures is too small to read clearly. Please increase font size for readability.

Response to Reviewer:

Thank you for pointing out the issue regarding the font size in several of the figures. We have carefully reviewed your suggestion and have increased the font size across all relevant figures (For example, Figures 2, 4, 5, and 6) to enhance readability. This includes adjustments to axis labels, legends, and all textual annotations to ensure they are clearly visible and legible in both print and digital formats.

The updated figures have been incorporated into the revised manuscript. We believe these changes significantly improve the clarity and visual accessibility of the figures, thereby enhancing the overall presentation of the results.

We sincerely appreciate your constructive feedback, which has helped improve the quality of the manuscript.

  1. The violin plots and Sankey diagrams are dense and may overwhelm readers. Consider simplifying these visualizations or moving some to supplementary materials.

Response to Reviewer:
Thank you for your valuable feedback regarding the complexity and readability of the violin plots and Sankey diagrams. In response to your suggestions, we have implemented the following changes to improve clarity and presentation:

  1. Violin Plot Adjustment:
    The original violin plot in Figure 2A was indeed visually dense. To improve readability and reduce complexity in the main text, we have relocated this plot to the supplementary materials (now presented as Supplementary Figure 1). This allows readers to access the full detail if desired, while streamlining the visual flow of the main manuscript.
  2. Sankey Diagram Enhancements:
    We have increased the font size of the labels in the Sankey diagram to enhance legibility. Additionally, we made minor layout adjustments to improve the diagram's clarity and overall visual balance, ensuring it is more accessible and easier to interpret.

We believe these revisions significantly enhance the usability and interpretability of the figures without compromising the depth of information conveyed.

Thank you again for your helpful suggestion, which has contributed to improving the overall quality of the manuscript.

Overall Response to Reviewer 1:

We sincerely thank the reviewer for your thorough, constructive, and insightful comments, which have significantly strengthened the clarity, rigor, and presentation of our manuscript. In response to your suggestions, we have made extensive revisions throughout the text, including (1) enhancing the transparency and statistical consistency of reported results (e.g., effect sizes, confidence intervals, and p-values), (2) improving the interpretability and design of key visualizations (violin plots, Sankey diagrams, and font sizing), (3) addressing potential confounders such as disease heterogeneity, sample imbalance, and data quality in both our machine learning and causal inference analyses, and (4) clarifying the rationale and biological interpretation behind our modeling strategies and key findings, including the treatment of protective genes in ALS and the hypothesis-generating nature of drug-target predictions.

We have also explicitly acknowledged limitations (e.g., tissue source of eQTLs, exclusion of APOE, lack of experimental validation) and clearly stated the scope of the study as a data-driven, exploratory framework. These refinements are reflected in the revised manuscript and supplementary materials. We greatly appreciate your efforts in helping us improve the manuscript and hope that the revised version meets your expectations.

Reviewer 2 Report

Comments and Suggestions for Authors

The authors primarily employ a series of computational approaches—including machine learning, Mendelian randomization, and molecular docking—to propose potential pathogenic mechanisms and candidate targets. However, it must be emphasized that the complete lack of experimental validation renders the conclusions of this study purely predictive, with severely limited scientific significance and practical value.

All key findings in this study are based solely on inference and simulation, without providing even the most basic level of biological validation (e.g., qPCR, Western blot, or cellular assays). While the computational framework may demonstrate a degree of technical integration, in scientific research, proposing “potential mechanisms” and “putative activities” based solely on computational predictions is far from sufficient. Without experimental support, such predictive results cannot constitute a meaningful scientific contribution.

If the authors are unable to provide experimental validation, it is strongly recommended that they explicitly position this work as a preliminary exploration or hypothesis-generating study. Furthermore, the conclusion section should be revised to moderate the strength of the claims, avoiding definitive terms such as “establishing mechanisms” or “identifying targets.”

Author Response

Reviewer 2

The authors primarily employ a series of computational approaches—including machine learning, Mendelian randomization, and molecular docking—to propose potential pathogenic mechanisms and candidate targets. However, it must be emphasized that the complete lack of experimental validation renders the conclusions of this study purely predictive, with severely limited scientific significance and practical value.

All key findings in this study are based solely on inference and simulation, without providing even the most basic level of biological validation (e.g., qPCR, Western blot, or cellular assays). While the computational framework may demonstrate a degree of technical integration, in scientific research, proposing “potential mechanisms” and “putative activities” based solely on computational predictions is far from sufficient. Without experimental support, such predictive results cannot constitute a meaningful scientific contribution.

If the authors are unable to provide experimental validation, it is strongly recommended that they explicitly position this work as a preliminary exploration or hypothesis-generating study. Furthermore, the conclusion section should be revised to moderate the strength of the claims, avoiding definitive terms such as “establishing mechanisms” or “identifying targets.”

Response to Reviewer:

Thank you for your thoughtful comment regarding the lack of experimental validation in our study. We greatly appreciate your perspective on the importance of experimental verification in scientific research.

We fully acknowledge that experimental validation is essential for confirming the computational predictions made in this study. However, as you rightly point out, this work primarily focuses on computational approaches, including machine learning, Mendelian randomization (MR), and molecular docking, to propose potential pathogenic mechanisms and candidate therapeutic targets. In the context of bioinformatics research, the goal of this type of analysis is often to generate hypotheses, identify promising targets, and suggest potential mechanisms that can be explored in future experimental studies.

  1. Positioning the Study in the Context of Bioinformatics Approaches to Biomedicine:

This study is positioned within the Topical Collection of “bioinformatics approaches to biomedicine”, where computational tools are used to analyze large-scale genetic, transcriptomic, and molecular data to gain insights into disease mechanisms and therapeutic possibilities. In this context, computational predictions are a valuable starting point for hypothesis generation. By leveraging existing biological datasets and advanced analytical techniques, we aim to identify genes, pathways, and molecular interactions that merit further experimental investigation. While the results are predictive, they are not meant to be conclusive but serve to guide future studies that could test these hypotheses experimentally.

Bioinformatics-based approaches are often the first step in understanding complex diseases such as neurological disorders, where direct experimental studies are challenging due to the heterogeneity and complexity of the diseases. For instance, machine learning and MR offer the potential to identify associations that may be difficult to discern using traditional experimental methods alone. The findings presented here, while computational, provide a valuable roadmap for experimentalists who can design targeted studies to validate and further investigate these predictions.

  1. Moderating the Strength of Claims:

We agree with your suggestion that the conclusions in the manuscript need to be moderated, particularly regarding the strength of claims. In response to your comment, we have revised the Abstract, Introduction, Discussion and Conclusion section etc. to clearly indicate that the findings are preliminary and hypothesis-generating, with an emphasis on the need for experimental validation. We have reframed the conclusions to avoid terms like “establishing mechanisms” and “identifying targets,” and instead highlight that the study presents potential mechanisms and candidate targets that warrant further experimental investigation.

Revised Conclusion:

In conclusion, this study proposes a multi-layered, computationally derived frame-work that highlights potential mitochondrial contributions to the pathogenesis of a broad spectrum of neurological diseases. While general mitochondrial bioenergetics appear con-sistently involved, our findings suggest that specific functional nodessuch as the meta-bolic checkpoint regulated by PDK1 or the crosstalk between mitochondrial genetics and immune signalingmay represent key points of vulnerability. Importantly, all conclu-sions are based on in silico methods, including MR, machine learning, and molecular docking, without experimental validation. Therefore, this work should be interpreted as a hypothesis-generating effort rather than a definitive mapping of causal mechanisms. The observed associations between neurological disease risk and peripheral blood phenotypes, such as red blood cell or platelet indices, support the possibility of a systemic brain-periphery mitochondrial axis. However, the functional relevance of these correla-tions requires further experimental substantiation. Similarly, the predicted interaction be-tween Celecoxib and PDK1, while suggestive of repurposing potential, remains specula-tive and must be confirmed through biochemical or pharmacological studies before thera-peutic inferences can be drawn. Taken together, this study offers a preliminary but inte-grative framework to prioritize candidate genes, pathways, and drug interactions for fol-low-up research. By combining diverse computational tools, it lays the groundwork for future biological investigations aimed at elucidating mitochondrial dysfunction in neuro-degeneration and developing more precise diagnostics and therapeutics."

We believe this revised conclusion accurately conveys the nature of the study and aligns with the expectations for bioinformatics research in biomedicine.

Final Thoughts:

We recognize the critical role of experimental validation in scientific research and view our bioinformatics study as a foundation for hypothesis-driven experimental work. By generating actionable insights, we aim to facilitate collaborations with experimental researchers to validate these findings, ultimately reducing the experimental burden. We believe these revisions address your concerns and clarify the study’s scope. Thank you for your constructive feedback, which has significantly improved the manuscript.

Round 2

Reviewer 2 Report

Comments and Suggestions for Authors

As I pointed out in my previous review comments, it is difficult for me, as a reviewer, to be fully convinced by a manuscript that relies solely on computational predictions without experimental validation. Although the computational methods employed by the authors appear technically sound, the conclusions drawn remain uncertain and less persuasive in the absence of supporting experimental evidence. I recommend that the authors strengthen the experimental validation in future work to enhance the credibility and academic impact of their study.

Round 2

As I pointed out in my previous review comments, it is difficult for me, as a reviewer, to be fully convinced by a manuscript that relies solely on computational predictions without experimental validation. Although the computational methods employed by the authors appear technically sound, the conclusions drawn remain uncertain and less persuasive in the absence of supporting experimental evidence. I recommend that the authors strengthen the experimental validation in future work to enhance the credibility and academic impact of their study.

Response:

We sincerely thank the reviewer for reiterating this important concern. As noted, our study adopts a computational, hypothesis-generating approach that integrates machine learning, Mendelian randomization, colocalization analysis, and molecular docking to explore potential mitochondrial mechanisms and therapeutic candidates across Alzheimer’s Disease (AD), Amyotrophic Lateral Sclerosis (ALS), Multiple Sclerosis (MS), and Parkinson’s Disease (PD).

We fully recognize that conclusions based solely on computational predictions are inherently limited and cannot substitute for experimental validation. While our analyses suggest possible associations—such as the implication of genes like PDK1, PGS1, and SLC25A38, and predicted interactions between Celecoxib and PDK1 (docking score: −6.52 kcal/mol)—we acknowledge that these results should be interpreted as preliminary and exploratory in nature.

In response to the reviewer’s suggestion, we have revised the Limitations section of the manuscript to clearly emphasize this point. As described in Lines 1092–1101 on Page 28, we now explicitly state the need for experimental confirmation and outline our proposed future directions. These include in vitro and in vivo studies, such as CRISPR/Cas9-based gene perturbation, Seahorse metabolic flux assays, and pharmacodynamic profiling in disease-relevant cellular and animal models.

We greatly appreciate the reviewer’s emphasis on scientific rigor and agree that experimental follow-up will be critical to substantiate and refine the hypotheses proposed in this study. We hope that our current work can serve as a starting point to guide future functional investigations.

Round 3

Reviewer 2 Report

Comments and Suggestions for Authors

As I mentioned in my previous comments, I have pointed out the limitations of this manuscript primarily from an experimental perspective. While the theoretical predictions presented by the authors appear to be methodologically sound, I personally believe that conclusions drawn solely from computational analysis without experimental validation are insufficient to support a complete scientific claim. That said, this is merely my personal viewpoint. At this stage, I have no further substantive suggestions for revision and find it difficult to make a clear recommendation for either acceptance or rejection. I suggest that the manuscript be further evaluated by a reviewer with specific expertise in theoretical or computational studies to better assess its scientific merit.